# Fascin-induced bundling protects actin filaments from disassembly by cofilin

Jahnavi Chikireddy[1], Léana Lengagne[1], Rémi Le Borgne[1], Catherine Durieu[1], Hugo Wioland[1], Guillaume Romet-Lemonne[1], and Antoine Jégou[1]

Actin filament turnover plays a central role in shaping actin networks, yet the feedback mechanism between network architecture and filament assembly dynamics remains unclear. The activity of ADF/cofilin, the main protein family responsible for filament disassembly, has been mainly studied at the single filament level. This study unveils that fascin, by crosslinking filaments into bundles, strongly slows down filament disassembly by cofilin. We show that this is due to a markedly slower initiation of the first cofilin clusters, which occurs up to 100-fold slower on large bundles compared with single filaments. In contrast, severing at cofilin cluster boundaries is unaffected by fascin bundling. After the formation of an initial cofilin cluster on a filament within a bundle, we observed the local removal of fascin. Notably, the formation of cofilin clusters on adjacent filaments is highly enhanced, locally. We propose that this interfilament cooperativity arises from the local propagation of the cofilin-induced change in helicity from one filament to the other filaments of the bundle. Overall, taking into account all the above reactions, we reveal that fascin crosslinking slows down the disassembly of actin filaments by cofilin. These findings highlight the important role played by crosslinkers in tuning actin network turnover by modulating the activity of other regulatory proteins.

## Introduction

Cells assemble a variety of actin filament networks to perform many fundamental cellular functions (Chalut and Paluch, 2016). Importantly, actin filament turnover needs to be tightly controlled for these networks to be functional and to timely adapt to external chemical and mechanical cues (Lappalainen et al., 2022; Blanchoin et al., 2014). Typically, actin filaments that polymerize in lamellipodia or the actin cortex are renewed within a few seconds (Fritzsche et al., 2013; Lai et al., 2008), while filaments composing stress fibers are stable over several minutes (Campbell and Knight, 2007; Saito et al., 2022; Valencia et al., 2021).

These actin networks are exposed to disassembly factors, among which proteins of the ADF/cofilin family are the main players (Svitkina and Borisy, 1999; Hotulainen et al., 2005). The activity of ADF/cofilin (hereafter cofilin) has been extensively characterized at the single actin filament level. It is known to induce actin filament fragmentation (Carlier et al., 1997; Blanchoin and Pollard, 1999), as well as filament depolymerization from both ends (McGough et al., 1997; McCullough et al., 2008; Suarez et al., 2011; Wioland et al., 2017; Schramm et al., 2017). Cofilin-induced severing is a complex mechanism that involves several reaction steps. Cofilin binds preferentially to ADP-actin filament segments rather than to "younger" ADP-Pi-rich segments (Maciver et al., 1991; Carlier et al., 1997; Suarez

et al., 2011). The binding of cofilin is cooperative, as cofilin molecules bind with a higher affinity to a site adjacent to a cofilin-occupied site along a filament, leading to the initiation of cofilin clusters that grow symmetrically in both directions (De La Cruz, 2005; Wioland et al., 2017). Importantly, cofilin binding locally shortens the helical pitch of the actin filaments (Galkin et al., 2001). Filament severing occurs at cofilin cluster boundaries, and four times more frequently at the boundary on the pointed end side of the cluster (Suarez et al., 2011; Gressin et al., 2015; Wioland et al., 2017). Recent CryoEM observations have revealed, with unprecedented detail, that the change of filament conformations upon cofilin binding does not propagate more than one actin subunit away from the cofilin boundary (Huehn et al., 2020). Reaction rates associated with the formation, growth, and severing of cofilin clusters have been measured for the three mammalian ADF/cofilin isoforms on α-skeletal and cytoplasmic β- and γ-actin filaments (Wioland et al., 2017, 2019a). Moreover, the ability of cofilin to apply torque on twist-constrained filaments increases the severing rate by up to two orders of magnitude (Wioland et al., 2019b). This result strongly suggests that the geometrical constraints imposed by crosslinkers in filament networks should enhance the ability of cofilin to disassemble these networks.

[1]Université Paris Cité, CNRS, Institut Jacques Monod, Paris, France.

Correspondence to Antoine Jégou: antoine.jegou@ijm.fr; Guillaume Romet-Lemonne: romet@ijm.fr.

In cells, actin filaments can be crosslinked by a large array of proteins that vary in size, affinity for the side of filaments, and the way they decorate filaments (Blanchoin et al., 2014; Gallop, 2019; Rajan et al., 2023). As a consequence, filament crosslinkers are major regulators of actin networks, specifically tuning their architecture and their mechanical properties (Claessens et al., 2006; Ma and Berro, 2018; Freedman et al., 2019; Banerjee et al., 2020; Lieleg et al., 2010). One special case of filament crosslinking is the parallel bundling of filaments. Such actin filament bundles are observed in filopodia and microspikes that emerge from the front of lamellipodia (Vignjevic et al., 2006), as well as in microvilli and stereocilia.

Previous studies have revealed that fascin is the main filament crosslinker in filopodia and microspikes (Adams and Schwartz, 2000; Vignjevic et al., 2006; Faix et al., 2009; Jacquemet et al., 2019; Damiano-Guercio et al., 2020). Fascin is a monomeric protein that arranges parallel filaments into bundles, with a hexagonal packing (Aramaki et al., 2016; Shin et al., 2009). In these bundles, fascin proteins each bind to two filaments and are positioned every half-pitch (36 nm) along the filaments (Edwards et al., 1995; Ishikawa et al., 2003; Jansen et al., 2011; Aramaki et al., 2016). This undertwists the actin filaments by 1° (Shin et al., 2009). Fascin in these bundles quickly turns over (Aratyn et al., 2007; Winkelman et al., 2016; Suzuki et al., 2020). Bundles formed by fascin reach a maximum size of typically 15–20 filaments both in vivo and in vitro (Jansen et al., 2011; Breitsprecher et al., 2011; Aramaki et al., 2016; Atherton et al., 2022; Hylton et al., 2022). This limit in size is thought to arise from the constraint of aligning the fascin binding sites along the actin filaments (Claessens et al., 2008).

So far, to our knowledge, only one study has addressed in vitro the impact of fascin bundling on cofilin-induced disassembly (Breitsprecher et al., 2011). Using both bulk pyrene and TIRF microscopy assays, Breitsprecher and colleagues proposed that fascin, by preventing the relaxation of the cofilin-induced torque, favored filament severing by cofilin. However, the absence of direct visualization of cofilin binding and severing events on filament bundles prompted us to further investigate the molecular details of the disassembly of fascin-induced filament bundles by cofilin.

Here, using purified proteins, we show that cofilin binds and fragments filament bundles slower than single actin filaments (Figs. 1 and 2). We report that the rates of initiation and growth of cofilin clusters on fascin-induced filament bundles are strongly reduced compared with what is observed on single filaments (Figs. 3 and 4). We further show that fascin is removed as cofilin clusters expand (Fig. 4). Strikingly, we reveal and quantify an inter-filament cooperativity mechanism where, after the creation of a first cofilin cluster, the subsequent initiation of cofilin clusters on adjacent filaments within the bundle is highly enhanced (Fig. 5). Numerical simulations integrating all the cofilin reaction rates determined experimentally recapitulate the observed fragmentation of both twist-unconstrained and twist-constrained actin filament bundles (Fig. 6). These observations led us to propose that filament crosslinking allows the cofilin-induced local change of helicity of one filament to be transmitted to other filaments (Fig. 6). Overall, our in vitro results provide novel molecular insights into how cofilin-induced actin network

disassembly is affected by filament crosslinking, a situation frequently encountered in cells.

## Results

All experiments were performed at 25°C, in a buffer at pH 7.4, with 50 mM KCl, 1 mM MgCl$_2$, using rabbit α-skeletal actin, recombinantly expressed mouse cofilin-1 and human fascin-1 (fascin, from here on) (see Materials and methods).

### Fascin-induced bundles are protected from cofilin binding

To investigate the overall activity of cofilin on fascin-induced filament bundles, we first performed experiments in so-called "open chambers," in a buffer supplemented with 0.2% methylcellulose (see Materials and methods). Actin filaments were polymerized from surface-anchored spectrin-actin seeds and exposed to 200 nM fascin as they grew. This fascin concentration allowed adjacent filaments to rapidly form stable bundles (Video 1), reaching on average 10 (±5, SD) filaments per bundle, as estimated per their actin fluorescence intensity. Actin filaments were then exposed to actin at critical concentration (~0.15 μM) and 200 nM fascin for 15 min so that filaments remained bundled and maintained a constant length as they became ADP-actin filaments. Upon addition of 80 nM mCherry-cofilin1, cofilin clusters appeared only very sparsely along filament bundles and the network did not disassemble over the course of 10 min (Fig. 1 A). In contrast, in control experiments without fascin, single actin filaments were readily targeted by cofilin, leading to their rapid fragmentation and disassembly (Video 2).

By quantifying the increase in cofilin fluorescence, normalized by the F-actin fluorescence, we observed that the binding of cofilin was ~15-fold slower on fascin-induced bundles than on single actin filaments (Fig. 1, B and C). Interestingly, larger bundles recruit less cofilin than smaller ones (~2.5 times less cofilin bound per F-actin for bundles of 10 (±4) filaments than of 3 (±0.8) filaments after 8 min, Fig. S1).

### Cofilin fragments two-filament bundles more slowly than single filaments

To investigate the molecular mechanisms of actin filament bundle disassembly by cofilin in more controlled conditions, we performed experiments using microfluidics (Jégou et al., 2011; Wioland et al., 2022). We first sought to quantify cofilin activity on two-filament bundles, the smallest bundle unit that can be assembled by fascin (Fig. 2 A and Video 3). Briefly, actin filaments were elongated from randomly positioned spectrin–actin seeds anchored on the glass surface of a microfluidic chamber to reach typically 10 μm in length. Upon exposure to 200 nM fascin, filaments bundled together, forming a majority of two-filament bundles, as revealed by their actin fluorescence intensity. Larger bundles were discarded from the analysis for this type of assay. Filaments that were far from the adjacent filaments could not form bundles, thereby providing a reference population of single filaments. Filaments were aged for 15 min to become ADP-actin filaments in the presence of fascin and actin. We verified that fascin crosslinking did not appear to significantly slow down phosphate release in actin filaments (Fig. S2),

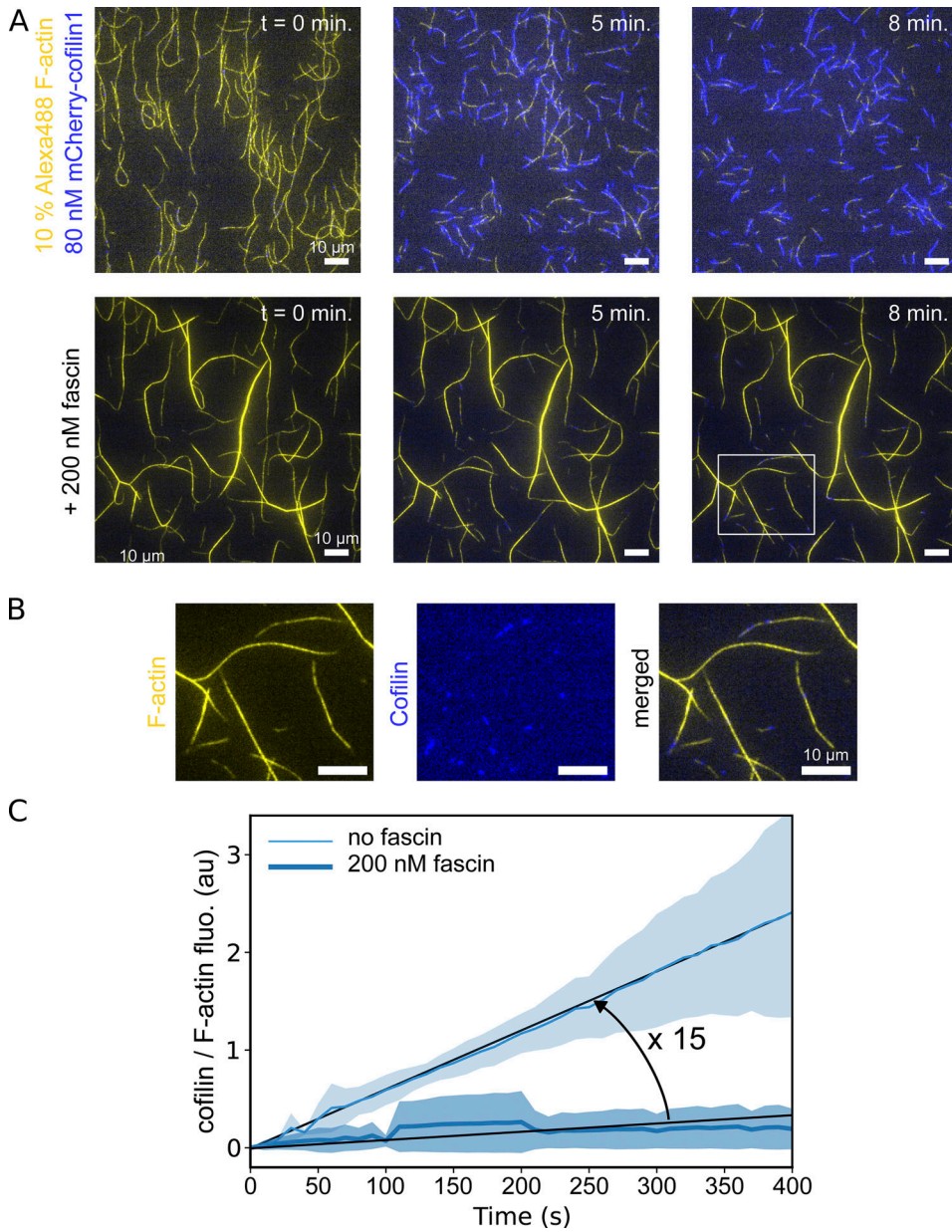

Figure 1. **Fascin-induced actin filament bundling slows down the recruitment of cofilin. (A)** 10% Alexa-488 labeled actin filaments, grown and aged for 15 min from surface-anchored seeds, in the absence or presence of 200 nM human-fascin1, in a buffer containing 0.2% methylcellulose, are subsequently exposed to 80 nM mCherry-cofilin1 at time 0 (see Materials and methods). **(B)** Fluorescence images showing the different channels from the white box shown in A, in the presence of fascin. **(C)** Fluorescence intensity of bound cofilin, normalized by the amount of F-actin, as a function of time. Both curves increase roughly linearly. Linear fitting indicates that cofilin binding to single actin filaments is 15 times faster than on fascin-induced actin filament bundles (N = 3 independent experiments). Shaded areas represent SD.

which otherwise would have affected cofilin binding on bundles compared with single filaments. Both bundles and single filaments were then exposed to 200 nM mCherry-cofilin-1 in the presence of fascin and actin to maintain filament bundling and filament length. We quantified the fraction of intact 5-μm long segments for single actin filaments and for two-filament bundles as a function of time upon exposure to cofilin. We observed a ~ninefold slower fragmentation for two-filament bundles compared with single filaments (Fig. 2, B and C).

To specifically investigate cofilin-induced fragmentation on larger bundles, we grew several actin filaments from spectrin-actin seeds attached to micrometer-size beads (in average, 10 filaments per bundle, Fig. S4 A; and Fig. 2, D and E). Cofilin-induced fragmentation on those large bundles was strongly reduced, by 40-fold compared with single actin filaments present in the same chamber (Fig. 2 F).

Fragmentation of a two-filament bundle requires the severing of two colocalized cofilin clusters, one on each filament, facing each other (Fig. 2 F). On its own, this constraint contributes to making the fragmentation of a filament bundle slower than that of a single filament. However, this does not explain why cofilin decoration appears to be slower on bundles than on single

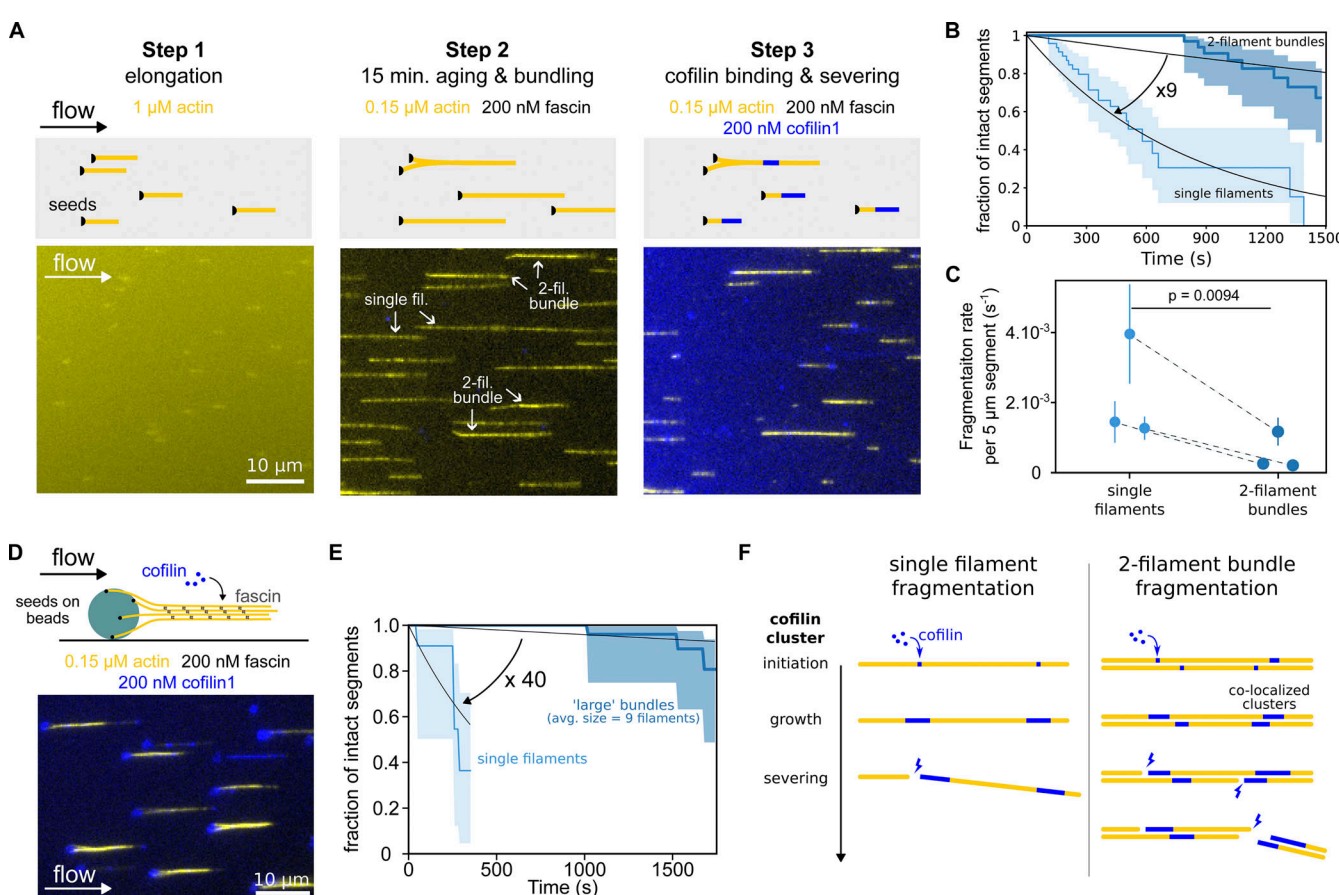

**Figure 2. Cofilin fragments fascin-induced bundles slower than single filaments. (A)** Schematics of the sequential steps to investigate fascin-induced two-filament bundle fragmentation by cofilin. Inside a microfluidic chamber, actin filaments were grown from randomly positioned surface-anchored seeds and aligned by the flow (step 1). Once elongated, filaments were aged and allowed to form two-filament bundles in the presence of fascin and actin for 15 min (step 2). They are then exposed to cofilin, actin, and fascin (step 3). Note that isolated filaments did not bundle and were used as reference single filaments for side-by-side comparison. **(B)** Result from a typical experiment showing the fragmentation, over time, of single actin filaments ($n$ = 53) and two-filament bundles ($n$ = 47) upon exposure to 200 nM cofilin, 0.15 μM actin, and 200 nM fascin. 95% confidence intervals are shown as shaded surfaces. There is an approximately ninefold difference in the rate of decay of those two populations, as obtained by single exponential fits (lines). **(C)** Fragmentation rates of single actin filaments and two-filament bundles when exposed to 200 nM cofilin. Rates are obtained from exponential fits as shown in B. Dashed lines indicate paired data points from single filament and bundle populations acquired simultaneously in the same microfluidics chamber ($N$ = 3 independent experiments; $n$ = 53, 21, 29 single filaments; $n$ = 47, 21, 30 two-filament bundles). Rates and error bars are obtained from exponential fits as shown in B. **(D)** Actin filaments were polymerized from spectrin–actin seeds adsorbed onto micron-sized glass beads to create larger bundles in an experiment otherwise similar to the one shown in A. Note that non-productive spectrin–actin seeds on beads are targeted by cofilin (blue), as revealed by the cofilin fluorescence appearing on the surface of the bead. **(E)** Survival fractions of intact 5-μm long segments of single actin filaments ($n$ = 12) or filament bundles ($n$ = 30, average size 9.3 (±3.2) filaments per bundle), upon exposure to 200 nM cofilin and 200 nM fascin, as a function of time. 95% confidence intervals are shown as shaded surfaces. There is a ~40-fold difference in the rates at which these two populations decrease, as obtained by single exponential fits (lines). **(F)** Schematics of the reactions that lead to cofilin-induced severing. The fragmentation of single filaments (left) results from the severing of one cofilin cluster, whereas the fragmentation of two-filament bundles (right) requires the severing of two "co-localized" cofilin clusters, one on each filament.

filaments (Fig. 1). Further, one may expect the severing rate per cofilin cluster to be faster in bundles than single filaments because of potential constraints on the twist imposed by filament bundling (Wioland et al., 2019b; Breitsprecher et al., 2011). To understand the mechanism responsible for the slower fragmentation of bundles, we thus decided to quantify the impact of bundling on the different cofilin molecular reactions that lead to fragmentation.

### Initiation of cofilin clusters is slower on actin filaments bundled by fascin
To better understand how cofilin disassembles filament bundles, we next sought to quantify the initiation of cofilin clusters,

which is the first step in cofilin-induced filament disassembly. First, we measured the impact of fascin on the initiation of cofilin clusters on single filaments. While fluorescently labeled fascin is easily detected on filament bundles (Aratyn et al., 2007; Winkelman et al., 2016; Suzuki et al., 2020), we could not detect the presence of fascin on single actin filaments, even at micromolar concentrations, as previously reported (Winkelman et al., 2016; Suzuki et al., 2020). Nonetheless, exposing single actin filaments to increasing fascin concentrations gradually decreased the initiation rate of cofilin clusters by up to ~twofold in the presence of 1 μM fascin compared with the absence of fascin (Fig. S3). This observation is consistent with the low affinity of

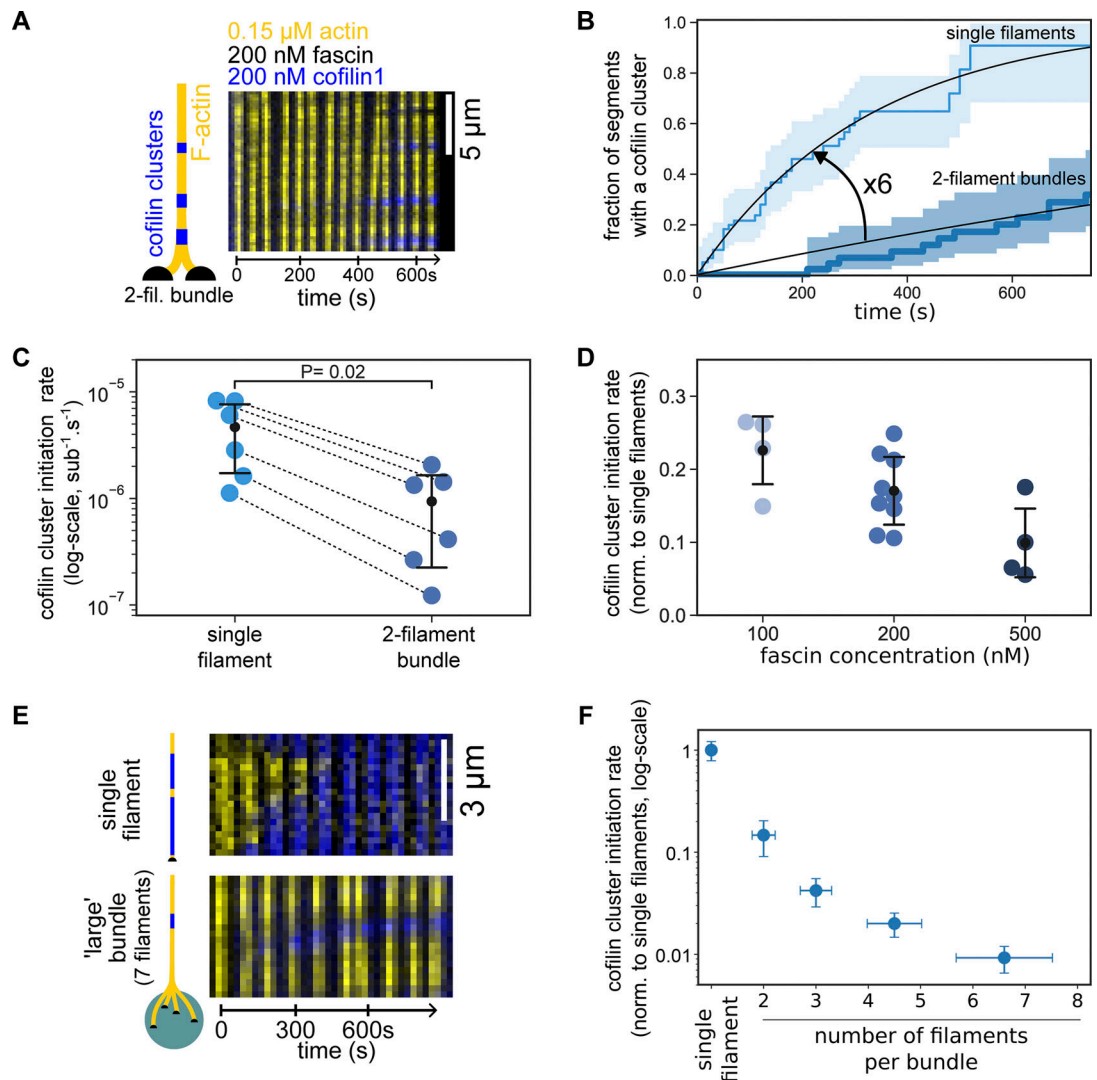

Figure 3. **Cofilin cluster initiation is slowed down by fascin-induced filament bundling. (A)** Time-lapse images of a two-filament bundle (yellow) showing the initiation of cofilin clusters (blue). **(B)** The fraction of 5-µm segments of single filaments (light blue, $n$ = 63 segments) or 2.5-µm segments of two-filament bundles (dark blue, $n$ = 47 segments) with at least one cofilin cluster, over time. 95% confidence intervals are shown as shaded surfaces. Black lines are single exponential fits. **(C)** The rate of initiation of cofilin clusters (log-scale), per cofilin binding site along actin filaments, measured from six independent experiments ($n$ > 30 segments for each population), in the presence of 200 nM cofilin, 200 nM fascin, and 0.15 µM actin. For each condition, the black dot represents the average, and the error bars represent the SD of the distribution. Rates were obtained from exponential fits as shown in B. Dashed lines indicate paired data points from populations acquired simultaneously in the same microchamber. The paired data shows consistently a sixfold difference in the cofilin cluster initiation rate. The P value is from a paired $t$ test. **(D)** Impact of fascin concentration on the initiation rate of cofilin clusters on two-filament bundles, normalized by the rate on single filaments ($n$ = 4, 9, and 4 independent experiments at 100, 200, and 500 nM fascin respectively, with >20 segments analyzed for each experiment). Rates were derived as shown in B. All conditions with 200 nM cofilin and 0.15 µM actin. For each fascin concentration, the error bar represents the SD of the distribution. **(E)** Timelapse images showing the appearance of cofilin clusters (blue) on either a single actin filament or a seven-filament bundle (yellow) grown from a micrometer-size glass bead, in conditions similar to A. **(F)** Impact of bundle size on the initiation rate of cofilin clusters (per cofilin binding site), normalized by the rate on single filaments, when exposed to 200 nM cofilin, 200 nM fascin and 0.15 µM actin ($N$ = 1 experiment, with 22, 10, 20, 40, and 25 segments analyzed for single filaments, and bundles of size 2, 3, 4–5, and 5–8 filaments, respectively). Horizontal error bars are SD for bundle size, and vertical error bars are the SD of the normalized cofilin initiation rates.

fascin for the side of single actin filaments. Furthermore, this indicates that cofilin and fascin may have overlapping binding sites, or that a more complex competition may exist between the two proteins, where the binding of one protein would induce conformational changes on neighboring actin subunits affecting the binding of the other protein. At 200 nM fascin, our reference fascin concentration in this study, cofilin binding to single filaments was not measurably affected.

In spite of having two filaments, thus twice as many potential binding sites, cofilin clusters form more slowly on two-filament bundles than on single filaments. Normalized by the number of binding sites, the rate of the initiation of cofilin clusters per binding site was reduced sixfold on two-filament bundles compared with single actin filaments, in the presence of 200 nM fascin (Fig. 3, A–C). This strong reduction in the initiation rate thus appears to play a key role in protecting bundles from cofilin-induced fragmentation.

JCB header

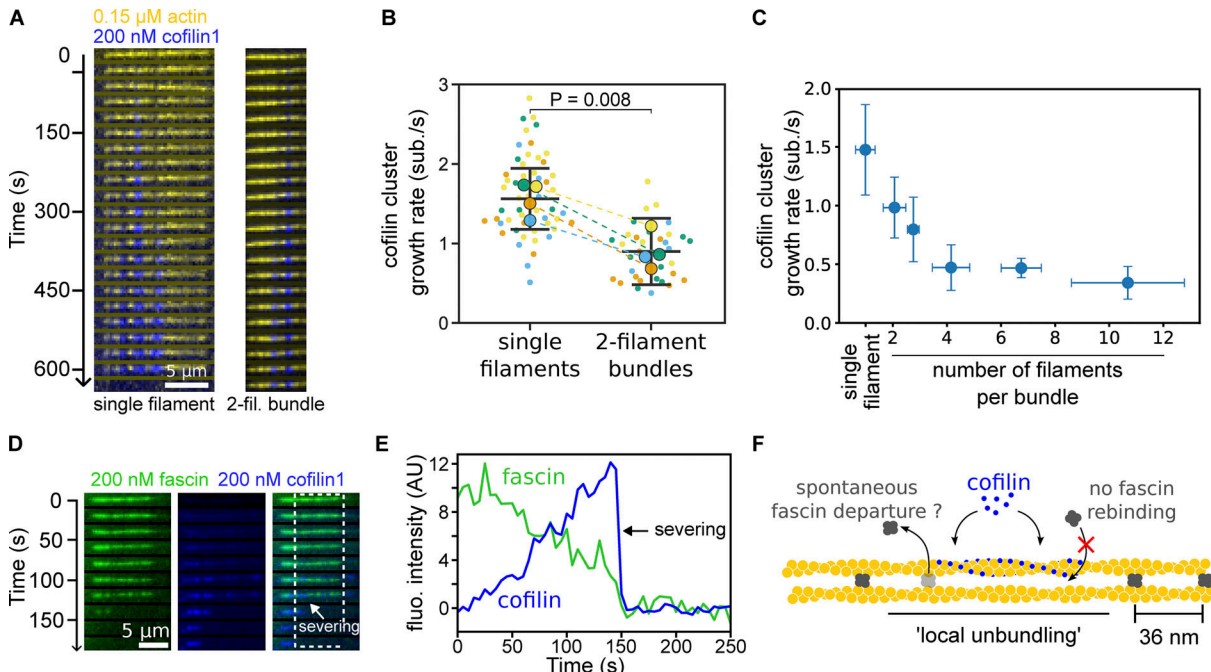

Figure 4. **Cofilin and fascin compete to bind filaments in bundles. (A)** Time-lapse images showing the growth of a cofilin cluster (blue) on a two-filament bundle (yellow). **(B)** Cofilin cluster growth rates on single filaments and two-filament bundles, in the presence of 200 nM cofilin, 200 nM fascin, and 0.15 µM Alexa488(10%)-G-actin (N = 4 repeats, each shown in a different color, with at least 10 cofilin clusters observed in each condition). Small data points are individual measurements (one per cluster) and the large points are the averages per repeat. Dashed lines indicate paired averages from populations acquired simultaneously in the same microchamber. The P value is from a paired *t* test. **(C)** Cofilin cluster growth rates as a function of bundle size (n = 11, 4, 6, 7, and 9 cofilin clusters analyzed on single filaments and bundles of an average size of 2, 3, 4, 7, and 11 filaments, respectively). Error bars are SD. **(D)** Time-lapse images showing a two-filament bundle with Alexa568-fascin (green) and exposed to eGFP-cofilin1 (blue). **(E)** Fluorescence intensity of fascin and cofilin, integrated over the region shown in D (dashed rectangle), as a function of time. **(F)** Schematics of the irreversible departure of fascin, caused by a cofilin cluster growing on a two-filament bundle.

We observed that cofilin clusters were homogeneously distributed along the bundle, excluding any effect of the mild tension gradient applied to the filament bundles by the microfluidics flow (force range 0–0.5 pN, Fig. S4). Higher fascin concentrations further decreased the cofilin cluster initiation rate on two-filament bundles by up to 10-fold at 500 nM fascin compared with single actin filaments (Fig. 3 D). At saturating fascin concentrations, i.e., above 500 nM (Winkelman et al., 2016; Suzuki et al., 2020), fascin binds every 13 actin subunits (i.e., every half-pitch of the actin filament) between two actin filaments (Jansen et al., 2011; Aramaki et al., 2016), leaving potentially 12 cofilin binding sites available between two bound fascins on each filament. Thus, the observed 6- to 10-fold reduction in the rate of initiation of cofilin clusters cannot be explained by a simple competition between cofilin and fascin to bind on actin filaments. Rather, the slower rate of initiation of clusters may originate from the slight change of filament helicity imposed by fascin bundling (Shin et al., 2009; Claessens et al., 2008).

We observed that the initiation rate of cofilin clusters, per cofilin binding site, strongly decreased with increasing bundle size (Fig. 3 F). The effect is so strong that cofilin clusters appear more slowly on large bundles (>5 filaments) than on single filaments in spite of their larger number of cofilin-binding sites (Fig. S5). Overall, these results consistently show that fascin-induced bundling strongly decreases the rate of initiation of cofilin clusters.

**Cofilin cluster growth is slowed down by fascin-induced bundling**
How fast cofilin clusters grow and decorate actin filaments is an important aspect of cofilin-induced disassembly. Thus, we next sought to measure the cofilin cluster growth rate on fascin-induced filament bundles. We observed that cofilin clusters grew at a 1.5-fold lower rate on two-filament bundles than on single filaments in the presence of 200 nM fascin (Fig. 4, A and B). This observation is consistent with the slower initiation of cofilin clusters reported above, although the effect appears to be much milder for cluster growth. Surprisingly, increasing fascin concentration did not appreciably decrease the cofilin cluster growth rate further (Fig. S6). Cofilin cluster growth rate decreased with increasing bundle size and seemed to plateau for large bundles (Fig. 4 C). Considering that actin filaments are arranged hexagonally in fascin-induced bundles, each filament can potentially be crosslinked to up to six filament neighbors, which would correspond to a maximum of six fascin proteins bound for every 13 actin subunits. Taken together, these observations seem consistent with cofilin binding being slowed down during the growth of a cofilin cluster in a bundle by the density of fascin bound along the filaments. This density depends mainly on the number of adjacent filaments and only marginally on the concentration of fascin in solution, in the range we used.

Cofilin binds subunits of actin filaments stoichiometrically (Galkin et al., 2001). We thus asked whether cofilin-saturated

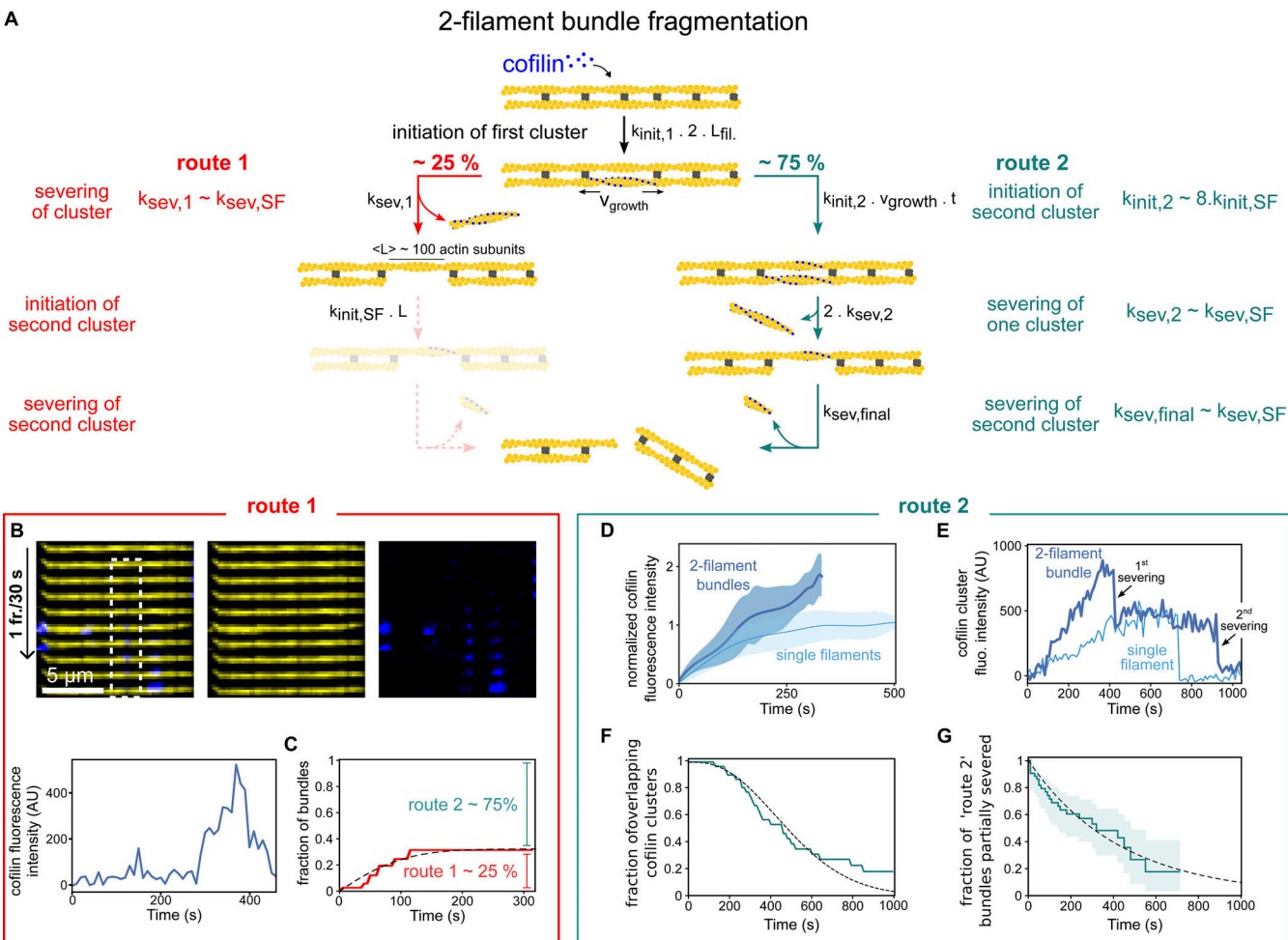

Figure 5. **Interfilament cooperativity leads to bundle fragmentation. (A)** Schematic representation of the two routes leading to the fragmentation of a two-filament actin bundle, after the first cofilin cluster has appeared, at a rate $k_{init,1}.2L_{fil}$, where $L_{fil}$ is the segment length of each filament of the bundle. For route 1, the initial cofilin cluster severs at its two boundaries (see main text), at a rate $k_{sev,1}$, before another cofilin cluster is initiated on the other filament of the bundle, in the region facing the first cofilin cluster. Subsequently, the initiation of a cluster on the remaining filament, followed by its severing, leads to bundle fragmentation, but these two final steps were never observed experimentally (shaded steps). For route 2, a cofilin cluster forms on the second filament in the region facing the first cofilin cluster before the latter severs, leading to the presence of two clusters in the same region. The sequential severing of the two clusters fragments the bundle. **(B)** An example illustrating the route 1 type of events: a single cofilin cluster (blue) severs a region of one filament in a two-filament bundle, while no other cofilin cluster has formed on the adjacent filament. The graph shows the intensity of the cofilin cluster in the dashed rectangle, as a function of time. **(C)** Fraction of bundles for which the first cofilin cluster severs, leaving behind no detectable cofilin, as a function of time. The fit of the experimental curve (see Materials and methods) yields $k_{sev,1} = 3 \times 10^{-3}$ s$^{-1}$ and $k_{init,2} = 1 \times 10^{-4}$ sub$^{-1}$.s$^{-1}$ ($n = 71$ events). The dashed line shows the best fit, using a chi-square minimization procedure (see Materials and methods). Three additional experiment repeats are shown in Fig. S10. In total, $N = 4$ independent experiments, with $n = 101, 71, 17, 28$ events, yield the following average values of $k_{sev,1} = 1.5 \times 10^{-3}$ s$^{-1}$ and $k_{init,2} = 4.7 \times 10^{-5}$ sub$^{-1}$.s$^{-1}$. **(D)** Average fluorescence intensity of cofilin clusters over a 1 μm wide segment on single filaments (light blue, $n = 10$) and on two-filament bundles (dark blue, $n = 29$), as a function of time, in the presence of 200 nM fascin and 200 nM cofilin, normalized by the maximum fluorescence intensity on single filaments. **(E)** Examples of cofilin intensity on a single filament (light blue) and a two-filament bundle (dark blue) as a function of time. The drop in intensity for the single filament is accompanied by the fragmentation of the single filament. For the two-filament bundle, a drop in the intensity to a lower value reveals the existence of two "co-localized" cofilin clusters. The second drop in intensity is accompanied by the complete fragmentation of the bundle. **(F)** Fraction of colocalized cluster regions that have not yet had a severing event, versus time ($n = 31$ events). The dashed line shows the best fit by a computed curve (see Materials and methods), using a chi-square minimization procedure. Three experimental repeats, with $n = 31, 33, 39$ events each, yield an average $k_{sev,2} = 1.06 (\pm0.2).10^{-3}$ s$^{-1}$ ($\pm$ SD). **(G)** Fraction of cofilin clusters remaining after the first severing event and that have not yet had the second severing event, versus time ($n = 53$ events, pooled from four independent experiments). Fit of the experimental curve by a single exponential yields $k_{sev,final} = 2.25 (\pm0.6).10^{-3}$ s$^{-1}$ ($\pm$95% confidence interval).

regions could also contain fascin. We observed that fluorescently labeled fascin was gradually excluded from expanding cofilin clusters on two-filament bundles (Fig. 4, D and E). One possible explanation could be that, as fascin quickly turns over in filament bundles (Aratyn et al., 2007; Suzuki et al., 2020), the departure of fascin would free space for cofilin and allow clusters to grow. This process seems irreversible as increasing the concentration of fascin in solution did not slow down cofilin cluster growth further, as would be expected if both proteins were directly competing for the same or overlapping binding sites (Fig. S6). Importantly, the exclusion of fascin from a cofilin-saturated region causes filaments of the bundle

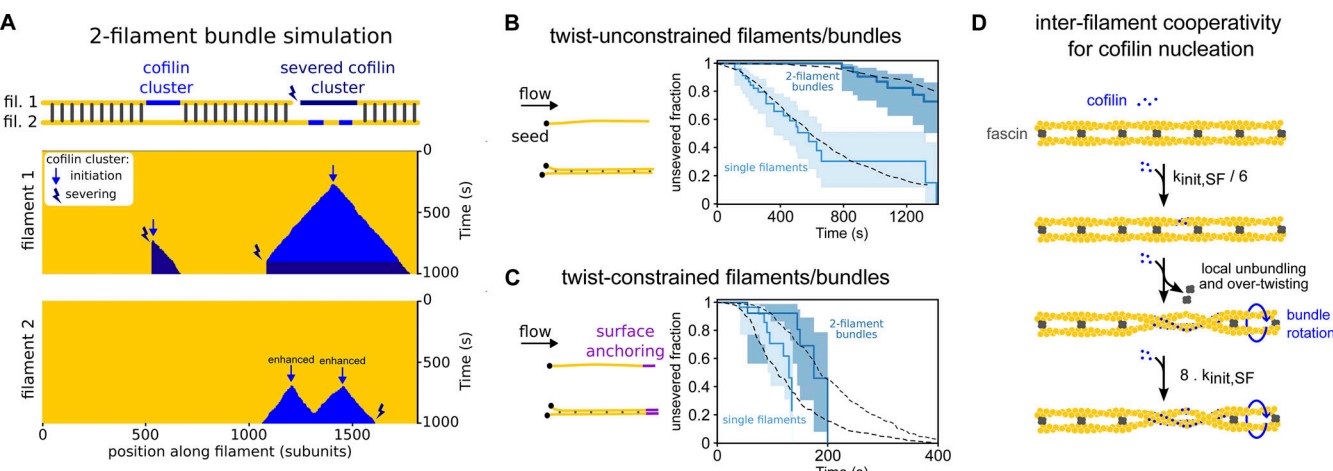

Figure 6. **Constraining two-filament bundles in twist highly enhances their fragmentation. (A)** Top: Schematics illustrating the numerical simulations of two 5-µm long actin filaments (yellow) interconnected by fascin (dark gray), where cofilin clusters (light blue) can form and sever filaments (dark blue). Bottom: Kymographs of two interconnected simulated filaments, showing cofilin cluster initiation (arrow) and severing (thunderbolt) events on each filament. The kymographs stop when the severing events on the two filaments result in the fragmentation of the bundle. Numerical values of reaction rates are summarized in Table S1. **(B and C)** Fraction of unsevered 5-µm segments that are (B) twist-unconstrained or (C) twist-constrained by being doubly attached to the glass surface, for single actin filaments (light blue, $n = 53, 34$ for filaments unconstrained and constrained in twist respectively) or two-filament bundles (dark blue, $n = 47, 16$ for bundles unconstrained and constrained in twist, respectively) upon exposure to 200 nM cofilin and 200 nM fascin, as a function of time. 95% confidence intervals are shown as shaded surfaces. Dashed lines correspond to the results obtained from numerically simulated segments ($n = 200$ for twist-unconstrained, 50 for twist-constrained segments), using experimentally determined rates and considering no inter-filament cooperativity in twist-constrained bundles (see main text). **(D)** Schematics of the interfilament cooperative twisting model. For a fascin-induced two-filament bundle, a first cofilin cluster on one of the filaments is initiated and starts growing, preventing fascin from binding locally. Local over-twisting caused by cofilin decoration is transmitted to the adjacent filament in the region devoid of fascin. This favors the binding of cofilin on the undecorated filament: the initiation rate of the second cofilin cluster is 48-fold higher than for the first cofilin cluster.

to be no longer crosslinked locally ("local unbundling," Fig. 4 F).

**Interfilament cooperativity in the initiation of cofilin clusters**
As fascin is locally excluded by the presence of a cofilin cluster on one of the two filaments of the bundle, we next sought to investigate whether this would favor the local initiation of a second cofilin cluster on the other filament. Plotting the local increase in fluorescence intensity of cofilin along two-filament bundles over time revealed that it often exceeded the intensity of individual cofilin clusters measured on single filaments (Fig. 5 D and Fig. S7). This observation indicates the presence of two co-localized cofilin clusters, one on each filament of the bundle.

To quantify the rate at which the second cofilin cluster is initiated, we must take into account all the reactions that together lead to the fragmentation of the two-filament bundle. After the initiation of a first cofilin cluster on either filament of a two-filament bundle, two different routes can lead to bundle fragmentation. They are presented schematically in Fig. 5 A. In the first route ("route 1"), the first cofilin cluster severs the actin filament it is bound to, at a rate $k_{sev1}$, before a cofilin cluster is initiated on the other filament in the region facing the first co-filin cluster. Alternatively ("route 2"), the second cofilin cluster is initiated before the first cluster severs its actin filament. In route 2, the probability of initiating a second cofilin cluster in the region where fascin has been excluded scales with the length of that region, which increases as the first cofilin cluster expands. The effective initiation rate of the second cofilin cluster can thus be written as $v_{cof,1} * t * k_{init,2}$, where $v_{cof,1}$ is the growth

velocity of the first cofilin cluster, $k_{init,2}$ is the cofilin cluster initiation rate per binding site in this fascin-free region for that cofilin concentration, and $t$ is time. The competition between the two reactions, the severing of the first filament versus the initiation of the second cluster, determines which route is followed by each two-filament bundle.

To estimate $k_{init,2}$, we sought to determine the relative importance of these two routes. One possibility is to identify severing events occurring on one of the two filaments within the bundle. The departure of one cofilin-saturated actin segment from the bundle, detectable as a drop in cofilin fluorescence, is a good surrogate for the severing of a cofilin cluster (see Materials and methods, Fig. S8).

We measured that ~25% (±8% SD, for a total of 217 bundles from 4 experiments) of the initial cofilin clusters fully severed and detached, leaving behind an unfragmented bundle with no detectable cofilin (Fig. 5, B and C). We interpreted these events as severing occurring before the initiation of a second cofilin cluster, thus corresponding to route 1. At every moment, the chosen route depends on the rate of each reaction, and the rate of route 2 increases with time as the first cofilin cluster grows. Numerically fitting the cumulative time distribution of events following route 1 (see Materials and methods) allowed us to simultaneously determine both rates $k_{sev,1}$ and $k_{init,2}$ (Fig. 5 C and Fig. S9). We obtained $k_{sev,1} = 1.5\ (\pm 1.2).10^{-3}\ s^{-1}$, which is comparable with the cofilin cluster severing rate measured on single filaments (rate $k_{sev,SF} = 2.1\ [\pm 0.3].10^{-3}\ s^{-1}$, Fig. S10). This indicates that the severing of cofilin clusters on two-filament bundles is not significantly affected by fascin crosslinking.

Remarkably, the initiation rate of the second cofilin cluster, $k_{init,2}$, is ~8 times higher than the initiation rate on single filaments ($k_{init,SF}$), thus 48 times higher than the initiation rate of the first cofilin clusters on two-filament bundles ($k_{init,1}$). This reveals the existence of a cofilin-driven "inter-filament cooperativity," where the presence of a cofilin cluster on a filament strongly favors the initiation of a cofilin cluster on the other filament of a two-filament bundle. We propose that this interfilament cooperativity arises from cofilin's ability to locally twist actin filaments, which would be transmitted, at least partially, to the adjacent filament in the fascin-free uncrosslinked region facing the first cofilin cluster. Such a mechanism would also relax the torsional stress induced by the cofilin cluster on the first filament, explaining why its severing rate is not enhanced as it would be if the filament was twist-constrained (Wioland et al., 2019b).

Following route 2, once there is a cofilin cluster on each filament of the bundle, each can sever at rate $k_{sev,2}$, and the first severing event thus occurs with an apparent rate $2*k_{sev,2}$. This translates into a drop in cofilin fluorescence intensity to a plateau of similar amplitude as for individual cofilin clusters on single filaments (Fig. 5 E). Fitting the cumulative time distribution of these events yields $k_{sev,2} \sim k_{sev,SF}$ (Fig. 5 F and Fig. S11; see Materials and methods). This indicates that the presence of two "side-by-side" cofilin clusters on a bundle does not affect the cofilin cluster severing rate. Ultimately, the severing of the remaining cofilin cluster occurs with a rate $k_{sev,final} \sim k_{sev,SF}$ (Fig. 5 G), as expected for an individual cofilin cluster on a single filament.

Could the cofilin-driven interfilament cooperativity we observed in two-filament bundles be also at play at larger bundles? Although cofilin cluster initiation events were rare on large bundles, we similarly could observe the initiation of colocalized cofilin clusters in larger bundles, as revealed by the cofilin fluorescence intensity (Fig. S12). On rare occasions, where large bundles fragmented completely (as quantified in Fig. 2 E), we could observe multiple steps in the decrease of cofilin fluorescence intensity, indicating multiple cofilin cluster severing events. However, the complexity of large bundles, with a large number of events occurring in parallel, prevented us from conducting an analysis similar to the one we performed on two-filament bundles.

To further investigate larger bundles, we imaged them using negative staining electron microscopy. In the absence of cofilin, filaments in bundles are arranged in a parallel manner, as previously reported in vitro (Jansen et al., 2011). Compared with the control, filament bundles exposed to cofilin show numerous sharp breaks (65 breaks per 122 μm of bundles, versus 4 breaks per 68 μm in the control (Fig. S13)). This is consistent with bundle fragmentation occurring at boundaries of colocalized cofilin clusters. Overall, it thus seems that large bundles also exhibit interfilament cooperativity for the initiation of cofilin clusters, which ultimately leads to bundle fragmentation.

**Initiation of the first cofilin cluster is the limiting factor in the fragmentation of bundles**

To recapitulate the impact of fascin-induced bundling on filament disassembly by cofilin, we performed Gillespie numerical simulations (Fig. 6 A), integrating all the individual reaction rates that we measured experimentally in microfluidics assays (Figs. 3, 4, and 5). First, we compared the fraction of unsevered two-filament bundles and unsevered single filaments as a function of time ($n$ = 200 simulated 5-μm long segments for each population). Without any free parameter, the simulated curves are in good agreement with the experimental data (Fig. 6 B). In the simulations, all two-filament bundle fragmentation events that occurred before 1,500 s were caused by the cooperative initiation of overlapping cofilin clusters (route 2 in Fig. 5), as experimentally observed. This indicates that, over these time scales, interfilament cooperativity is the dominant pathway leading to bundle fragmentation, and that the initiation of a new cluster where one filament has already severed (route 1 in Fig. 5) is extremely rare, explaining why we never observed these final steps in our in vitro experiments.

To assess the impact of the different steps leading to bundle fragmentation, we modified the reaction rates in our numerical model. We focused on the two key effects of fascin-induced bundling: the hindered initiation of the first cofilin cluster (low $k_{init,1}$), which delays the fragmentation of the bundle, and interfilament cooperativity for the initiation of the second cofilin cluster (high $k_{init,2}$), which favors the fragmentation of the bundle. We first simulated the situation where cofilin cluster initiation is unaffected by fascin crosslinking (i.e., $k_{init,1} = k_{init,2} = k_{init,SF}$), keeping all the other rates unchanged. In this situation, there is no hindrance to the first cluster initiation and no interfilament cooperativity. This resulted in an approximately twofold faster bundle fragmentation compared with what we observed experimentally (Fig. S14). We then simulated the situation where the first cluster initiation is hindered, but where there is no interfilament cooperativity (i.e., $k_{init,1} = k_{init,2} = 1/6\ k_{init,SF}$). In this situation, bundle fragmentation is approximately twofold slower than in our experiments (Fig. S14). These two alternative simulated scenarios indicate that both the delayed initiation of the initial cofilin clusters and the interfilament cooperativity of the following cofilin clusters strongly impact the rate at which two-filament bundles are fragmented by cofilin.

**Twist-constrained bundle fragmentation**

When single actin filaments are constrained in twist, by being attached to fixed anchors at each end, cofilin severing is dramatically increased due to the inability of filaments to relax cofilin-induced torsional stress (Wioland et al., 2019b). In cells, filopodia making contacts with the extracellular matrix or other cells could constrain the twist of the fascin-induced actin filament bundles they contain (Jacquemet et al., 2015). We thus sought to quantify the cofilin-induced fragmentation of twist-constrained fascin-induced two-filament bundles. To do so, the downstream part of filament bundles was anchored to the glass surface using biotin–streptavidin linkages (Fig. 6 C). We observed that twist-constrained bundles fragmented significantly faster than unanchored ones (Fig. S15). We observed that the initiation rate of cofilin clusters was similar for both twist-constrained and twist-unconstrained fascin bundles (Fig. S16), in agreement with observations on single actin filaments (Wioland et al., 2019b).

The rapid fragmentation of twist-constrained two-filament bundles prevented us from directly quantifying the initiation rate of the subsequent cofilin clusters that overlapped with the initial ones, as well as cluster growth and severing rates. To assess if twist-constrained bundle fragmentation resulted from interfilament cooperative cofilin cluster initiation, we performed numerical simulations where the growth of cofilin clusters creates a torque and increases the severing rates of cofilin clusters. As above, numerical simulations were performed using parameters determined experimentally (Fig. 3, 4, and 5) and with a cofilin-induced severing rate that increases exponentially as torque accumulates, as previously characterized for single twist-constrained filaments (Wioland et al., 2019b) (Fig. 6 C). In simulations with an interfilament cooperative cofilin initiation rate of the same amplitude as the one determined for unconstrained bundles ($k_{init,2} = 8\ k_{init,SF}$), simulated twist-constrained bundles fragmented appreciably faster than experimentally observed (Fig. S17). Simulations performed without interfilament cooperative initiation ($k_{init,2} = k_{init,SF}$) appear in better agreement with our experimental observations (Fig. 6 C). In this latter case, 75% of the first cofilin clusters severed before a second overlapping cofilin cluster could be initiated (route 1).

Overall, these results indicate that constraining the twist of fascin-induced filament bundles abolishes the interfilament cooperative initiation of the second cofilin clusters. Constraining the twist also prevents the relaxation of the mechanical torque induced by the growth of cofilin clusters and accelerates bundle fragmentation.

From our kinetic analysis, we propose the following model that recapitulates the binding of cofilin to fascin-induced two-filament bundles (Fig. 6 D). Initially, actin filaments in fascin-induced bundles are in conformations that are less favorable for cofilin binding than isolated actin filaments. Once a cofilin cluster appears, its expansion locally triggers fascin unbinding and prevents it from rebinding. The increase of filament helicity induced by cofilin causes a local twisting of the entire bundle, thereby changing the helicity of the adjacent filament in the fascin-free region facing the cofilin cluster. In this region, the increase in filament helicity enhances cofilin affinity and thus locally promotes the initiation of a cofilin cluster (interfilament cooperativity).

## Discussion

In this study, we reveal that actin filaments are severed more slowly by cofilin when they are bundled by fascin, and we decipher the molecular details underlying this phenomenon. In particular, we show that bundle fragmentation is slower than what would occur if each filament in the bundle behaved as an individual filament and fragmentation of the bundle occurred upon the colocalization of independent severing events on each filament (Fig. 2 F). Strikingly, this is primarily due to the slower initiation of the first cofilin clusters on fascin-induced bundles compared with single filaments. In addition, one could have expected fascin to constrain filament twist, leading to the faster severing of filaments by cofilin (Wioland et al., 2019b;

Breitsprecher et al., 2011). We observed that this is not the case for two-filament bundles and appears unlikely for larger bundles as well. Indeed, we report here the exclusion of fascin from within cofilin clusters and a strong increase in the initiation of cofilin clusters on adjacent filaments. This interfilament cooperativity mechanism leads to the colocalized initiation of cofilin clusters and permits bundle fragmentation faster than if the initiation of cofilin clusters on adjacent filaments were purely random. To our knowledge, this is the first time such interfilament cooperativity is ever reported. To explain this mechanism, we propose that the cofilin-induced change of helicity produced locally on one filament can be transmitted to the adjacent filaments within the bundle (Fig. 6 D).

Overall, these observations show that fascin crosslinking strongly impacts cofilin activity, with two opposing effects: first, it hinders the initiation of the first cofilin clusters on cofilin-free regions of the bundles, and second, it greatly accelerates the initiation of additional cofilin clusters on bare filaments, adjacent to an already existing cluster. Globally, the first effect dominates, and fascin-induced bundles are protected from severing. This is especially the case for large bundles.

Our study provides new molecular insights into the initial binding of cofilin on intact fascin-induced bundles and the importance of bundle size. Previously, using pyrene-actin bulk experiments, Breitsprecher and colleagues reported diminished cofilin binding to fascin-induced filament bundles (Breitsprecher et al., 2011). In spite of this, their observation of fluorescently labeled actin filament bundles seemed to indicate an efficient severing activity. Since cofilin was not fluorescently labeled, they could not observe cofilin clusters and they proposed that severing was enhanced because fascin served as anchors along filaments and impeded cofilin-induced changes in filament helicity. Similar to this proposed mechanism, we have reported that cofilin severing is enhanced on artificially twist-constrained single actin filaments (Wioland et al., 2019b). Here, we show that this mechanism does not occur in fascin-induced bundles. In particular, we reveal that fascin is locally excluded from cofilin-decorated segments. This exclusion of fascin has important consequences as it may allow cofilin to locally change the helicity of filaments. In the case where filament bundles are anchored to the surface of the chamber, twist is constrained, cofilin-induced interfilament cooperativity is suppressed, and bundles fragment rapidly due to accelerated cofilin severing (Fig. 6). These observations indicate that the bundle's ability to rotate is an important factor, and they are consistent with our hypothesis that interfilament cooperativity is due to the local transmission of changes in helicity from the cofilin-decorated filament to its neighbors.

Why is cofilin binding hindered on bundles? Fascin sits between two bundled filaments every 36 nm (Aramaki et al., 2016; Jansen et al., 2011). Many actin subunits are therefore available for cofilin to bind between two consecutive bound fascins. Consequently, direct steric competition between fascin and cofilin cannot account for the sixfold reduction in initial cofilin binding on the filaments of a bundle. However, fascin-induced bundling moderately decreases the helicity of filaments by ~1° (Shin et al., 2009). One hypothesis explaining the slower

binding of cofilin could therefore be that, due to fascin-induced bundling, filaments are "trapped" in conformations that are less favorable for cofilin to bind. This interpretation complies with the notion of "filament breathing," where thermal fluctuations of the filament conformation (subunit conformations, inter-subunit interfaces, local filament curvature, twist, etc.) strongly modulate the binding of regulatory proteins (Galkin et al., 2010; Schramm et al., 2019; Reynolds et al., 2022). Here, fascin would decrease the time spent by actin filaments in conformations that are preferred for cofilin to bind. We have previously shown that the binding of cofilin to twist-constrained actin filaments is unaffected (Wioland et al., 2019b). We therefore argue that reducing the fluctuations of actin filaments has probably a stronger impact on cofilin binding than shifting the average helicity by ∼1° (Jégou and Romet-Lemonne, 2020).

Another important aspect of fascin-induced bundling is filament packing. As opposed to larger bundlers (e.g., alpha-actinin), fascin tightly packs filaments, with an interfilament distance of only 6 nm (Jansen et al., 2011). For freely diffusing proteins, such as cofilin, the accessibility of filaments that form the core of the bundles could thus be hindered. In addition, to reach the core, cofilin has to diffuse through a medium composed of absorbing obstacles (i.e., the outer actin filaments of the bundles), so cofilin could be locally depleted (Saxton, 1994; Manhart et al., 2019) and filaments at the core of bundles be "protected" from cofilin. We indeed show that cofilin binds less efficiently to larger bundles (Fig. 3). For larger bundles, one would have to wait for filaments in the periphery to first be decorated by cofilin to make filaments of the core accessible to cofilin. In conclusion, fascin-induced bundling could thus hinder cofilin binding through various mechanisms, whose relative contributions are difficult to disentangle.

In cells, fascin has been identified as the main crosslinker of actin filaments in filopodia (Vignjevic et al., 2006; Jacquemet et al., 2019). Our results indicate that fascin-induced bundles in filopodia may be protected from cofilin present in the cytoplasm, thanks to a slower initiation of cofilin clusters. When cofilin clusters are formed, they locally remove fascin from the bundle. Such a phenotype has been recently reported in the filopodia of neuronal growth cones, thanks to EM tomography. In one study, some filaments in large filopodial bundles displayed segments with a shorter helical pitch that appeared disconnected from the rest of the hexagonally packed bundle (Atherton et al., 2022). Another study, using live fluorescence microscopy in addition to EM, further revealed that sparse regions of filament bundles in filopodia were decorated by cofilin (Hylton et al., 2022). These regions coincided with filopodia kinks or wavy shapes that would correspond to portions of the bundles where filaments are less crosslinked by fascin and thus more prone to bending due to the presence of cofilin. These deformations of filopodia are also reminiscent of the clockwise rotation of filopodia (Tamada, 2019; Leijnse et al., 2022), which could thus be attributable to cofilin interfilament cooperativity inducing the rotation of actin filament bundles. Rotation of filopodia in cells has been proposed to be an essential aspect of the emergence of the left-right asymmetry during brain development (Tamada, 2019) and could thus also rely on cofilin. In

migrating cells, nascent focal adhesion sites connecting filopodia to the substrate probably constrain the twist of the actin bundles (Jacquemet et al., 2015). The fact that cofilin more rapidly disassembles twist-constrained bundles could thus enable a fast transition from a filopodia-like to a stress fiber actin structure that would connect to nascent focal adhesions.

Is the complete fragmentation of bundles happening in cells? While cofilin probably does sever filaments within bundles, the high concentration of profilin–actin in cells (Funk et al., 2019) may allow for the regrowth of severed filament-barbed ends along bundles, at least in cases where filament-barbed ends are not saturated by cofilin (Wioland et al., 2017). Re-elongating barbed ends may then be crosslinked "back" into the bundle, thanks to free fascin available in the cytosol (Vignjevic et al., 2006), and this would be a possible self-repairing mechanism. However, these newly generated barbed ends could just as easily be rapidly capped by capping proteins CP, which is also present in filopodia (Sinnar et al., 2014; Edwards et al., 2014). The final outcome would thus depend on the relative abundance of these regulatory proteins.

Apart from filament disassembly, we envision that cofilin could potentially play an unanticipated role by inducing the rotation of parts of actin networks, as discussed above in the case of bundles in filopodia. Here, we reveal that fascin crosslinking is a factor that modulates cofilin activity. This may also apply to networks made by other crosslinking proteins. For example, alpha-actinin, which forms dimers and thus constitutes a larger and more flexible crosslinker than fascin (Christensen et al., 2017), may differentially impact cofilin's ability to reorganize actin bundles. Interestingly, in fibroblasts, ADF/cofilin has been recently identified to contribute to the establishment of chiral radial stress fibers (Tee et al., 2023), for which alpha-actinin is thought to be the main crosslinker. Our results thus open new perspectives on how filament crosslinking contributes to the regulation of the turnover of actin networks.

## Materials and methods
### Biochemistry
#### Protein purification
α-skeletal muscle actin was purified from rabbit muscle acetone powder following the protocol described in Wioland et al. (2017), based on the original protocol from Spudich and Watt (1971).

Spectrin-actin seeds were purified from human erythrocytes as described in Wioland et al. (2017), based on the original protocol by Casella et al. (1986).

Human fascin-1 (Uniprot: Q16658): 6xHis-fascin1 was purified as described previously in Suzuki et al. (2020).

Mouse cofilin-1 (Uniprot: P18760): fluorescent fusion protein 6xHis-eGFP-cofilin-1 and 6xHis-mCherry-cofilin-1 were purified as described previously in Kremneva et al. (2014).

#### Protein labeling
Rabbit α-skeletal muscle actin was fluorescently labeled on accessible surface lysines of F-actin using Alexa-488 succinimidyl ester (ref. A20000; Life Technologies). To minimize the effects

of the fluorophore, we used a labeling fraction of 10% for both microfluidics and open-chamber assays.

Actin was similarly labeled with biotin using biotin succinimidyl ester (ref. A39256; Life Technologies).

Fascin was labeled on surface cysteines using Alexa-568 maleimide (ref. A20341; Life Technologies), leading to a labeling fraction of ~180%.

Cofilin-1 was fused with mCherry or eGFP at their N-terminus (Kremneva et al., 2014). We systematically used 100% labeled cofilin-1, for which activity has been verified previously (Wioland et al., 2017).

## Fluorescence microscopy
For both open chambers and microfluidics assays, we took care to minimize the impact of light exposure that affects cofilin binding to actin filaments.

We observed day-to-day variations in the activity of cofilin. To minimize this limitation, we took advantage of the microfluidics experiments: we exposed single filaments and filament bundles that are side-by-side at the surface of the chamber to the same cofilin conditions to derive fold-change between cofilin activities on single filaments and bundles.

## Buffers
All experiments were performed in F-buffer: 5 mM Tris HCl pH 7.4, 50 mM KCl, 1 mM MgCl$_2$, 0.2 mM EGTA, 0.2 mM ATP, 10 mM DTT, and 1 mM DABCO. The concentrations of DTT and DABCO were chosen to limit light-induced artifacts. Buffers were supplemented with 0.2% methylcellulose (4000 cP at 2%; M0512, Merck) for open chamber assays to keep filaments in the vicinity of the glass bottom and image them using TIRF microscopy (with a laser penetration depth ~80 nm).

## Cofilin binding assay in open chambers
Experiments were conducted in chambers made by assembling two 22 × 40 mm #1.5 coverslips, spaced by melted parafilm. The surface was incubated with 2 pM spectrin–actin seeds for 5 min and then passivated with bovine serum albumin (BSA, 50 mg/ml, for 5 min). Actin filaments were elongated from surface-anchored spectrin–actin seeds, using 10% labeled Alexa488-actin in F-buffer, supplemented with 0.2% methylcellulose. They were then aged to become fully ADP-actin filaments by exposing them to 0.15 µM 10% Alexa488-actin for 15 min in the presence or absence of 200 nM fascin. Filaments were then exposed to 80 nM mCherry-cofilin1 and 200 nM fascin. Note that the cofilin concentration used in open chambers is substantially lower than in microfluidics experiments due to the propensity of methylcellulose to increase protein concentration close to the glass surface.

## Bundles and single filaments in microfluidics
Microfluidics experiments were done with Poly-Dimethyl-Siloxane (PDMS, Sylgard) chambers based on the original protocol from Jégou et al. (2011) and described in detail in Wioland et al. (2022). Briefly, glass coverslips were previously cleaned in sequential ultrasound baths of 2% Hellmanex, 2 M KOH, pure water, and ethanol, each for 30 min, with extensive rinsing in

pure water between each step. A PDMS cross-shaped chamber with three inlets (inner main channel dimensions of 20 µm in height, 800 µm in width, and ~1 cm in length) was mounted onto a cleaned glass coverslip. Both PDMS and coverslip were previously plasma-activated for 30 s to allow them to bind tightly to each other. The microfluidics chamber was connected to solution reservoirs by blue PEEK tubings. Pressure in the reservoirs was controlled and flow rates were monitored using microfluidic devices MFCS-EZ and Flow Units (Fluigent).

We used spectrin-actin seeds to anchor filaments by their pointed end to the microfluidics coverslip surface: the surface was incubated with 2 pM spectrin–actin seeds for 5 min, then passivated with bovine serum albumin (BSA, 50 mg/ml, for 5 min). Filaments were then grown using a solution of 1 µM 10% Alexa488-actin in the F-buffer. Filaments were finally aged for at least 15 min by exposing them to a critical concentration of 0.15 µM 10% Alexa488-actin to ensure that the actin is >99.9% in ADP-state, in the presence or absence of 200 nM fascin. We verified that fascin bundling did not significantly slow down Pi release during the aging process (Fig. S2).

## Twist-constrained filaments and bundles
To constrain the twist of single filaments and bundles (Fig. 6), filaments were first grown from surface-anchored spectrin–actin seeds in a microfluidics chamber previously passivated by 50:1 BSA:biotin-BSA (0.5 mg/ml in F-buffer, for 5 min). Filaments were sequentially elongated, first using 1 µM 10% Alexa488-actin to grow ~10 µm-long segments, then using 0.5 µM 1% biotin-actin for 1 min to generate a ~2 µm-long biotinylated segment at their barbed end. Filaments were then aged for 15 min by exposing them to a critical concentration of 0.15 µM 10% Alexa488-actin to ensure that the actin is >99.9% in ADP-state, in the presence of 200 nM fascin. Bundles were subsequently exposed to neutravidin (3 µg/ml) in the presence of 200 nM fascin and 0.15 µM actin, for 5 min, to anchor their distal end to the biotinylated surface. Finally, bundles were exposed to 200 nM mCherry-cofilin-1, 200 nM fascin, and 0.15 µM 10% Alexa488-actin to quantify the initiation rate of cofilin clusters and the fragmentation of twist-constrained bundles and single filaments.

## Formation of larger bundles
To form fascin-induced bundles composed of more than two filaments, filaments were grown from non-specifically adsorbed 1-µm in diameter glass beads. Beads were previously functionalized with spectrin–actin seeds (100 pM in F-buffer for 5 min, rinsed twice in F-buffer). The coverslip surface was then passivated with BSA (50 mg/ml, for 5 min). The number of filaments per bundle was quantified using actin fluorescence prior to cofilin exposure (Fig. S5 A) using the fluorescence of single filaments that grew from the surface as a reference.

## Image acquisition
Experimental chambers were positioned on a Nikon TiE inverted microscope equipped with a 60× 1.49 NA oil immersion objective. In microfluidics, we systematically used the epifluorescence illumination mode to avoid fluorescence variation due to filament or bundle height in the evanescent/TIRF

illumination mode. For open chamber assays, methylcellulose strongly constrains filament and bundle height to ~50 nm above the glass surface, and TIRF illumination was used. 100 mW tunable lasers (iLAS2; Gataca Systems) were used for TIRF illumination. A 120 W Xcite Exacte lamp (Lumen Dynamics) was used for epifluorescence imaging. The TiE microscope was controlled either by Metamorph or ImageJ/Micromanager (Edelstein et al., 2014) software. Images were acquired by a sCMOS Orca-Flash4.0 V3.0 (Hamamatsu) camera. The temperature was controlled and set to 25°C (using an objective-collar heater, Oko-lab).

## Negative staining electron microscopy

Actin filament bundles are assembled by incubating 10 μM actin with 1 μM fascin in F-buffer for 30 min at room temperature. The solution was diluted 10× in F-buffer with 200 nM fascin in the presence or absence of 500 nM cofilin and allowed to react for 60 s. Three microliters were then deposited on a 400-mesh copper grid with a conventional glow discharge activated carbon film. After an additional 60 s, the sample was briefly drained by the Whatman paper filter, and the grid was covered with a drop of 1% (wt/vol) aqueous uranyl acetate for 30 s and then dried. The grids were examined at 120 kV with TEM (Tecnai 12; Thermo Fisher Scientific) equipped with a 4K CDD camera (Oneview, Gatan).

## Data analysis

For both single filaments and bundles, measurements were performed on 5-μm long segments. For single filaments, we excluded the region that extends two pixels away from the seed position due to the sharp bending of the filament. For bundles, as they are formed from filaments elongated from surface-anchored seeds that might be microns away from each other, only the central regions for which the actin fluorescence intensity clearly indicate that two filaments are bundled together were analyzed.

During the course of the experiment, severing events often occur at the seed location due to the sharp bend of the filament (Wioland et al., 2019b). All these events do not correspond to regular severing events and were taken into account as "censoring" events. We used the Kaplan–Meier method to construct survival fractions and estimate confidence intervals (Kaplan and Meier, 1958), which are implemented in the "lifelines" package in Python.

Cofilin cluster initiation on ADP–actin filaments or bundles was quantified by manually detecting the appearance of cofilin clusters giving rise to detectable growing clusters. The time-dependent cumulative distribution of the fraction of actin segments harboring at least one cofilin cluster is fitted by a single exponential to obtain the cofilin cluster initiation rate per cofilin-binding site per second.

Cofilin cluster growth rates were obtained using kymographs and by manually measuring the width of cofilin clusters for at least three different time points.

Cofilin cluster severing events and filament/bundle fragmentations were manually tracked and listed as time points to build "survival" curves and fitted as follows:

- The population of isolated cofilin clusters on two-filament bundles that severed before a second cofilin cluster could be detected (Fig. 5 C) was fitted using a chi-square minimization procedure (curve_fit function from the scipy package in Python). By simulating, using the Euler method, the outcome of a population of 200 cofilin clusters that grow at a rate $v_{growth}$ and can either sever with a rate $k_{sev,1}$ or cooperatively initiate a new cofilin cluster on the adjacent filament in the region that overlaps with the initial cofilin cluster at a rate $k_{init,2} * t * v_{growth}$, where $k_{init,2}$ is the rate of initiation of the second cofilin cluster. The least-square fitting procedure gives the most probable values for the two free parameters, $k_{sev,1}$ and $k_{init,2}$.

- The survival fraction of two-overlapping cofilin clusters on two-filament bundles (Fig. 5 F) was fitted using a chi-square minimization procedure (curve_fit function from the scipy package in python), by simulating, using the Euler method, the outcome of a population of 200 cofilin clusters that grow at a rate $v_{growth}$, allow the initiation, at a rate $k_{init,2}$, of a second cofilin cluster that overlaps with the first one and grows at the similar rate $v_{growth}$, and the two cofilin clusters sever at an effective rate $2 \times k_{sev,2}$. The least-square fitting procedure gives the most probable value for the only free parameter, $k_{sev,2}$.

- The survival fraction of the isolated cofilin clusters that arise from an initial population of two overlapping cofilin clusters on two-filament bundles, and where one of them has already severed (Fig. 5 G), is fitted by a single exponential decay function with a decay rate $k_{sev,final}$ as a free parameter.

- Single filament and bundle fragmentation were fitted by a single exponential decay function to derive a fold difference in the rate at which fragmentation occurs for both populations (Fig. 2).

## Impact of the detection of cofilin severing events occurring on filament bundles

The departure of one cofilin-saturated actin segment, which was detectable as a drop in cofilin fluorescence, occurs once it has severed at its two boundaries. We expect, in the vast majority of cases, that severing occurs first at the pointed end side of cofilin clusters (Suarez et al., 2011; Wioland et al., 2017). Because of our experimental configuration, the second severing event may then occur immediately after as, due to the flow, the dangling cofilin-decorated region may pivot around its barbed-end side where it is still bound by fascin to the second filament (Fig. S8). This would result in a sharp bend which would accelerate the severing at this boundary, as previously reported (Wioland et al., 2019b). In that case, the severing of the first cluster boundary would lead to an almost immediate detachment of the cofilin cluster from the bundle. We performed a specific assay, described in Fig. S8, allowing us to quantify that around half of the severing events occurring at the pointed end side of a cofilin cluster coincided with a drop in cofilin fluorescence (i.e., rapid departure of cofilin-actin segments).

These cofilin-actin segments will either sever at the opposite boundary before a cofilin cluster is initiated on the adjacent filament in the region facing the first cofilin cluster, and this will be counted as route 1 event, or a cofilin cluster will be initiated on the adjacent filament. In the latter case, the rate of initiation of the second cofilin cluster is the rate observed on single filaments, as there is no interfilament cooperativity (i.e., the first

cofilin cluster does not transmit any twist as it has severed at one boundary). In the competition between those two types of events, severing at the other boundary will dominate and occur ~90% of the time, considering the measured rates and a cluster size of typically ~100 subunits. Experimentally, these events will appear as route 1 events. The remaining 10% will "appear" as route 2 events as two cofilin clusters are overlapping and the severing of these two clusters will lead to bundle fragmentation, although they are not route 2 events per se. The error in distinguishing between route 1 and route 2 events is thus reasonably small (~10%) and does not appreciably impact the estimated rates.

Overall, the drop in cofilin fluorescence is thus a reasonable surrogate of the cofilin severing event.

## Numerical simulations

We performed Gillespie numerical simulations to simulate cofilin activity on single filaments (1 segment) or two-filament bundles (2 segments), each segment being 5 μm long (1818 actin subunits in length). Absolute parameters used in the simulations are reported in Table S1. We applied the following rules:

- A cofilin cluster forms at a rate:
  - On bare actin subunits of filaments of 2-filaments bundles: $k_{init,bundle}$.
  - On bare actin subunits of single filaments: $k_{init,SF} = 6 \times k_{init,bundle}$.
  - On actin subunits that face a cofilin-saturated actin region on the adjacent filament of two-filament bundles: $k_{init,bundleEnhanced} = 48 \times k_{init,bundle}$.
  - On actin subunits that face a cofilin-occupied actin region on the adjacent filament of two-filament bundles, and which at least one of the boundaries of this latter cluster has severed: $k_{init,SF}$.
- A cofilin cluster grows symmetrically toward both the pointed and barbed ends, at a rate $v_{growth}$, in all cases.
- A cofilin cluster severs at a rate $k_{sev}$, in all cases, 80% (resp. 20%) of the time toward the pointed end (resp. barbed end), as previously reported in Wioland et al. (2017).
- For twist-constrained single filaments or bundles, a cofilin cluster severs at a rate that is exponentially increased as a function of the cofilin-induced torque using $k_{sev,torque} = k_{sev} \times \exp(\alpha.\Gamma/k_B.T)$, where α is a constant whose value was set to 5, to match the previously best-reported value in Wioland et al. (2019b), Γ is the applied torque (in pN.nm) and $k_B.T$ is the product of the Boltzmann constant and the temperature (4.1 pN.nm, at 25°C). The torque was computed as in Wioland et al. (2019b) using $\Gamma = \nu.d\theta/dL.C_a.C_c/(\nu.C_a + (1 - \nu).C_c)$, with ν the cofilin density along the segment, dθ = 4.7° the rotation along the long filament axis induced by the binding of an additional cofilin molecule, dL = 2.7 nm, the length of actin subunits in the filament, and $C_a$ (resp. $C_c$) the torsional rigidity of a bare (resp. cofilin-decorated) single actin filament. We used previously reported values from Prochniewicz et al. (2005), where $C_a = 2.3 \times 10^3$ pN.nm²/rad and $C_c = 0.13 \times 10^3$ pN.nm²/rad.
- The simulation is halted, and the time of the event is recorded for analysis, if:

- For both single filaments and two-filament bundles: they are fully decorated by cofilin and no fragmentation has occurred, or the simulation time has reached $t_{max} = 3,000$ s (8,000 s, for Fig. S14).
- For single filaments: the filament is fragmented, i.e., upon the first cofilin cluster severing event.
- For two-filament bundles: when two (partially) severed cofilin-decorated regions, one on each filament, are colocalized. In this case, fascin cannot crosslink all the actin segments together.

## Softwares

All measurements (length, distance, fluorescence intensity, severing time, etc.) were performed manually on Fiji/ImageJ. Data analysis and statistical significance tests and Gillespie numerical simulations were done using Python (with numpy, scipy, panda, and lifelines packages).

## Statistical significance

A comparison of the survival distributions between the two samples was done using the P value from the log-rank test, using the lifelines package in Python.

The sample data means were compared using the Welch's paired two-samples $t$ test to derive a P value, using the "ttest_rel" function from the scipy package in Python. Superplots were generated using SuperPlotsOfData—a web app for the transparent display and quantitative comparison of continuous data from different conditions (Goedhart, 2021), available online at https://huygens.science.uva.nl/SuperPlotsOfData/.

## Online supplemental material

Fig. S1 shows that cofilin binds less efficiently to larger filament bundles. Fig. S2 shows that fascin crosslinking does not appear to slow down Pi release in actin filaments. Fig. S3 shows that fascin decreases the rate of cofilin cluster formation on single actin filaments. Fig. S4 shows that cofilin clusters appear homogeneously along two-filament bundles. Fig. S5 shows that the rate of initiation of cofilin clusters decreases with bundle size. Fig. S6 shows the impact of fascin on the growth rates of cofilin clusters. Fig. S7 shows cofilin fluorescence reveals the presence of overlapping cofilin clusters along two-filament bundles. Fig. S8 shows that the curvature of cofilin-actin segments induced by the flow may accelerate severing. Fig. S9 shows the fraction of cofilin clusters that will sever before a cofilin cluster is initiated on an adjacent filament of a two-filament bundle in the region facing the first cofilin cluster. Fig. S10 shows the cofilin cluster severing rate on single actin filaments. Fig. S11 shows the fraction of colocalized cofilin clusters leading to two-filament bundle fragmentation. Fig. S12 shows the observations of colocalized cofilin clusters on bundles composed of more than two filaments. Fig. S13 shows negative staining EM images of fascin-induced actin filaments exposed to cofilin. Fig. S14 shows numerical simulations indicating the impact of interfilament cooperativity for the initiation of cofilin clusters on the fragmentation of two-filament bundles. Fig. S15 shows that twist-constraining two-filament bundles lead to faster fragmentation by cofilin. Fig. S16 shows the initiation of cofilin clusters on twist-constrained filaments. Fig. S17 shows numerical simulations showing the impact of interfilament cooperativity for the initiation of cofilin

clusters on the fragmentation of twist-constrained two-filament bundles. Table S1 shows cofilin reaction rates on single and bundle actin filaments. Video 1 shows fascin-induced actin filament bundles exposed to cofilin, in an open chamber assay. Video 2 shows single actin filaments exposed to cofilin in an open chamber assay. Video 3 shows fascin-induced actin filament bundles exposed to cofilin in a microfluidics assay. Data S1 shows the data for all the plots shown in the figures of the article. Each figure panel corresponds to a different sheet.

### Data availability
The data underlying all figures are available in the published article and its online supplemental material (Data S1).

## Acknowledgments
We acknowledge the ImagoSeine core facility of the Institut Jacques Monod, a member of the France BioImaging infrastructure (ANR-10-INBS-04), and GIS-IBiSA.

J. Chikireddy, L. Lengagne, and A. Jégou were supported by the European Research Council (ERC) under the European Union's Horizon 2020 research and innovation program (grant agreement StG-679116 to A. Jégou). J. Chikireddy was also supported by the Labex WhoAmI? of Université Paris Cité. G. Romet-Lemonne was supported by Agence Nationale de la Recherche (grant RedoxActin).

For the purpose of Open Access, the authors have applied a CC BY public copyright license to any author-accepted manuscript version arising from this submission.

Author contributions: Conceptualization—A. Jégou and G. Romet-Lemonne. Data curation—J. Chikireddy, A. Jégou, and G. Romet-Lemonne. Formal analysis—J. Chikireddy, L. Lengagne, H. Wioland, G. Romet-Lemonne, and A. Jégou. Funding acquisition—G. Romet-Lemonne and A. Jégou. Investigation—J. Chikireddy, L. Lengagne, H. Wioland, R.L. Borgne, C. Durieu, G. Romet-Lemonne, and A. Jégou. Methodology—J. Chikireddy, H. Wioland, A. Jégou, and G. Romet-Lemonne. Project administration—G. Romet-Lemonne and A. Jégou. Resources—G. Romet-Lemonne. Software—A. Jégou. Supervision—G. Romet-Lemonne and A. Jégou. Validation—H. Wioland, G. Romet-Lemonne, and A. Jégou. Visualization—J. Chikireddy, A. Jégou, and G. Romet-Lemonne. Writing—original draft—J. Chikireddy, and A. Jégou. Writing—review & editing—A. Jégou and G. Romet-Lemonne.

Disclosures: The authors declare no competing interests exist.

Submitted: 20 December 2023

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

# Supplemental material

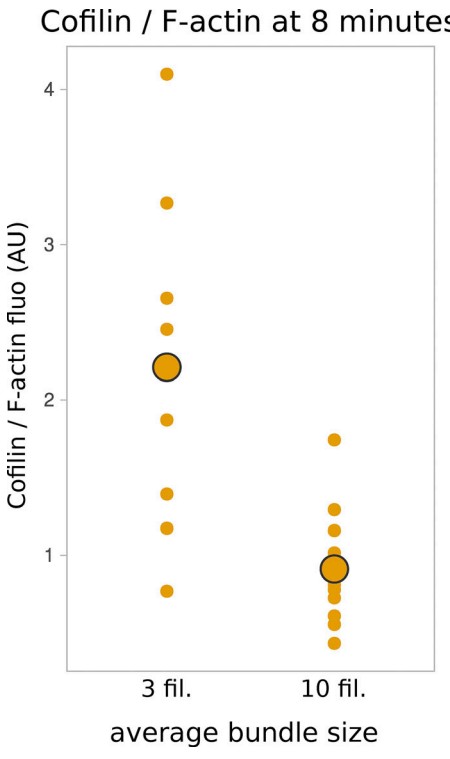

Figure S1.    **Cofilin binds less efficiently to larger filament bundles.** Fluorescence intensity of cofilin, normalized by the amount of F-actin, bound after 8 min after cofilin introduction in the "open chamber," on segments of typically 4 μm of fascin-induced filament bundles of average size 3 (±0.8, SD, *n* = 8 segments) and 10 (±4, *n* = 10 segments) filaments. The large dots represent the median value of each population.

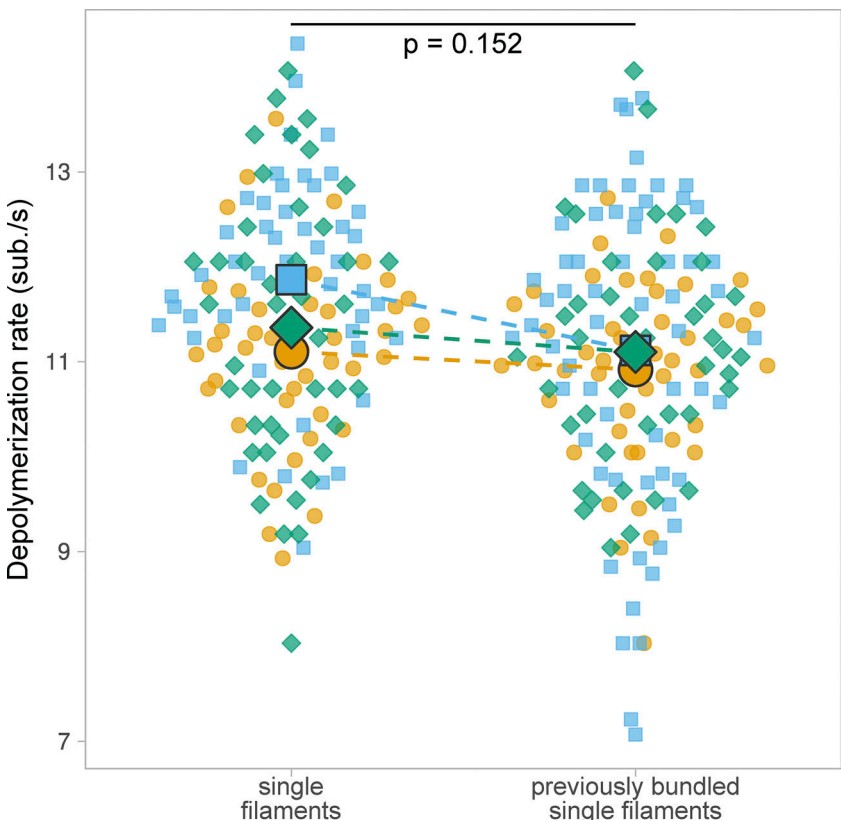

Figure S2. **Fascin crosslinking does not appear to slow down Pi release in actin filaments.** Actin filaments elongated from spectrin–actin seeds in microfluidics chambers were bundled and aged for 15 min by exposing them to 0.15 µM actin and 200 nM fascin. Actin filaments were then unbundled and they depolymerized as single filaments upon exposure to buffer only. In the absence of fascin in solution, bundled filaments become individual isolated filaments after typically 30 s. The depolymerization rates (measured over 3 min) of individual actin filaments that were initially either isolated filaments or part of two-filament bundles when exposed to fascin, were quantified ($N$ = 3 repeats, with $n$ = 44, 47, and 44 for single filaments, and $n$ = 45, 63, and 42 for two-filament bundles, respectively). Large symbols represent median values. The P value is calculated from the comparison of the paired median values.

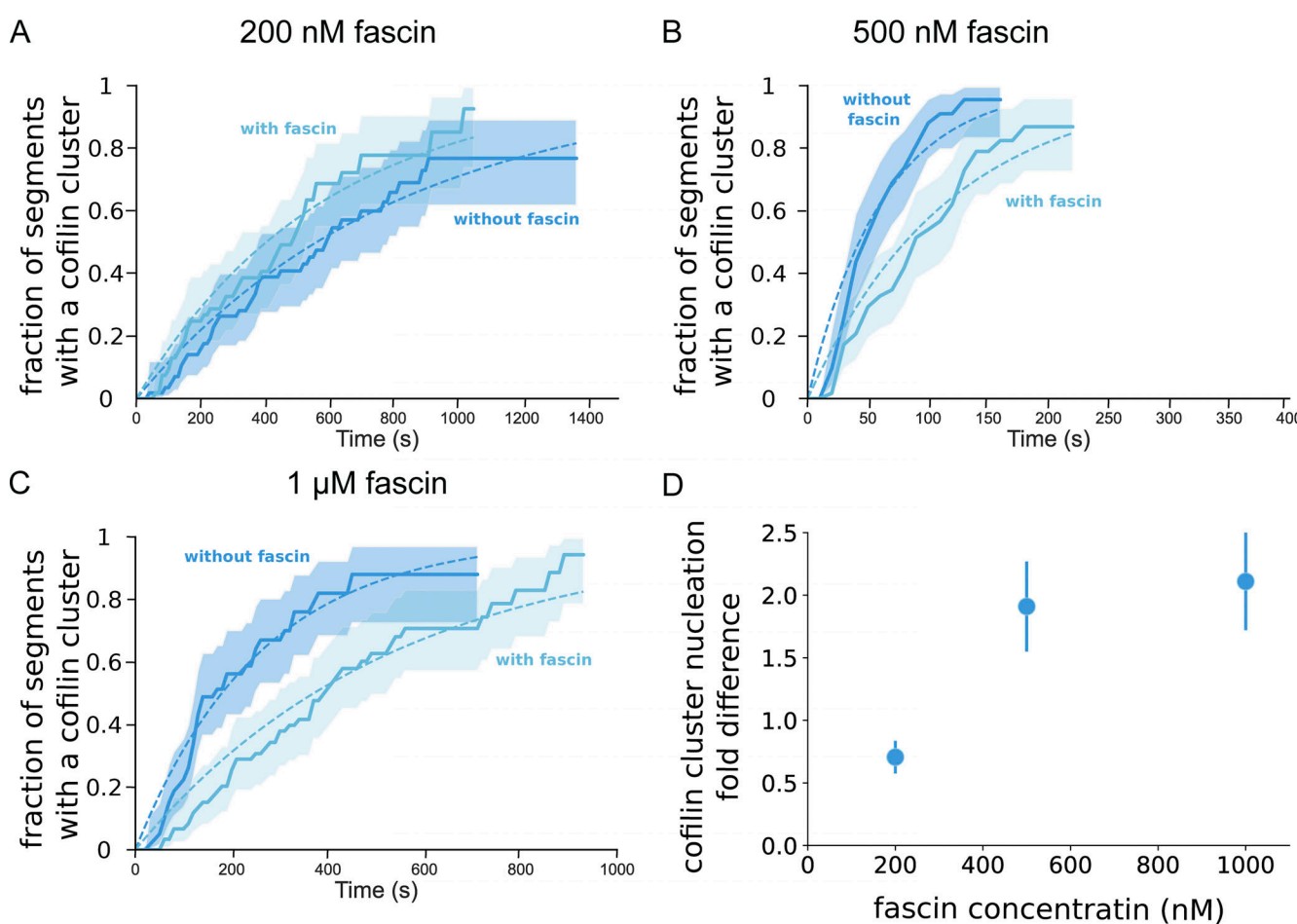

Figure S3.  **Fascin decreases the rate of cofilin cluster initiation on single actin filaments. (A–C)** Two side-by-side populations of single actin filaments are exposed in a microfluidics chamber to 200 nM cofilin, 0.15 µM actin, and either no or (A) 200 nM, (B) 500 nM, (C) 1 µM fascin (n = 54, 58, 60 segments with fascin, and n = 57, 53, 60 segments without fascin). Fraction of 5-µm segments of single actin filaments with at least one cofilin cluster, as a function of time. 95% confidence intervals are shown as shaded surfaces. Curves are fitted with a single exponential function to derive the cofilin cluster initiation rate for each population. **(D)** Cofilin cluster initiation rate fold difference between the population of single actin filaments exposed to fascin or not. Error bars for each condition are 95% confidence intervals, based on the sample sizes of the two survival fractions.

Figure S4. **Cofilin clusters appear homogeneously along two-filament bundles.** Cumulative distribution of the localization of *n* = 32 cofilin clusters along 7-µm two-filament bundles, when exposed to 200 nM mCherry-cofilin-1, 200 nM fascin, and 0.15 µM actin. The first micron of the bundle was excluded to avoid any effect due to curvature close to the anchorage point of the filaments. The straight dashed line represents the case where clusters would be perfectly homogeneously distributed.

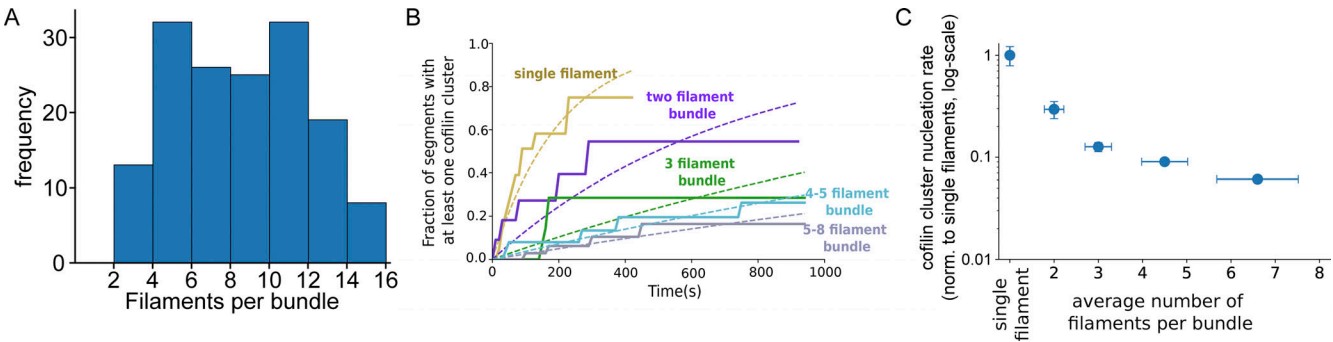

Figure S5. **The rate of initiation of cofilin clusters decreases with bundle size. (A)** Size distribution of fascin-induced bundles formed from actin filaments polymerized from individual beads in the presence of 200 nM fascin. Average size = 9.7 (±4.8) filaments per bundle, *n* = 179 bundles. Size is determined by the actin fluorescent intensity relative to the intensity of single actin filaments (*n* = 10 filaments). **(B)** Fraction of 5-µm segment filament bundles harboring at least one cofilin cluster, over time, as a function of bundle size (single filaments *n* = 22, 2-fil. *n* = 10, 3-fil. *n* = 20, 4–5-fil. *n* = 40 and 5–8-fil. bundles *n* = 25). Bundle sizes were determined based on the relative fluorescence intensity of actin compared to single filaments. **(C)** Impact of bundle size on the rate of appearance of cofilin clusters, normalized by the rate on single filaments and by the number of filaments in bundles. Values are obtained from exponential fits of curves shown in A. Error bars are SD.

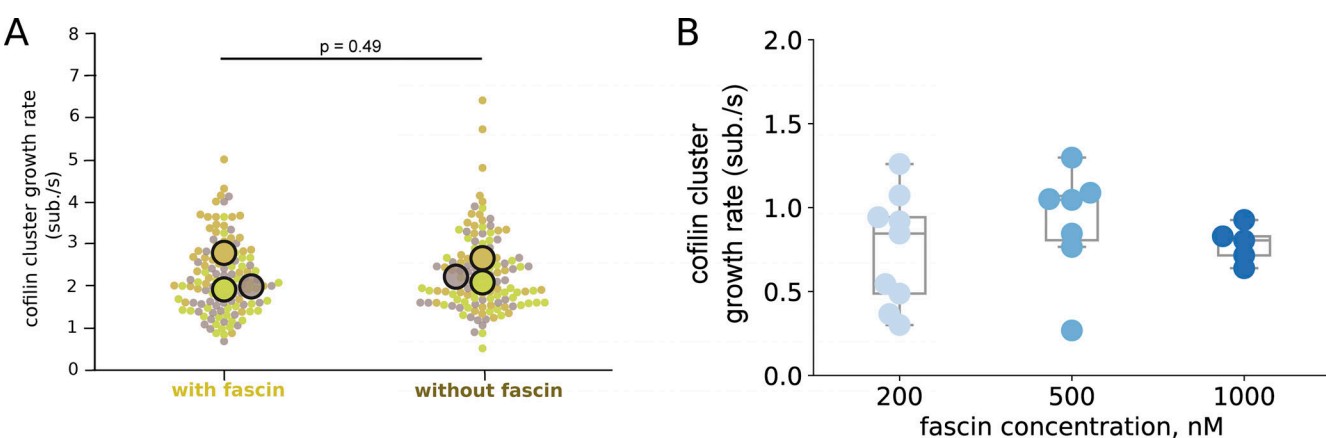

Figure S6. **Impact of fascin on the growth rates of cofilin clusters. (A)** Growth rates of cofilin clusters on single filaments exposed to 200 nM mCherry-cofilin1 in the presence or absence of 200 nM fascin, N = 3 experiments, n > 40 cofilin clusters for each experiment. Large symbols represent averages over individual measures from independent experiments. Paired *t* test P value = 0.494. **(B)** Growth rates of cofilin clusters on two-filament bundles exposed to 200 nM cofilin and various fascin concentrations (n = 9, 7, 5 cofilin clusters for 200, 500, and 1,000 nM fascin respectively). One-way ANOVA test P value = 0.56.

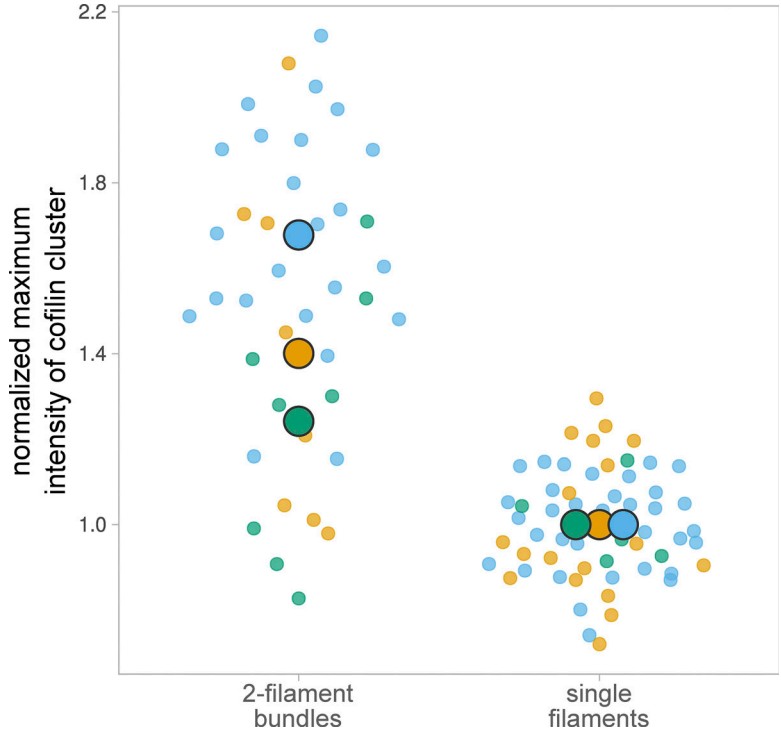

Figure S7. **Cofilin fluorescence reveals the presence of overlapping cofilin clusters along two-filament bundles.** Maximum fluorescence intensity of cofilin measured over 1 µm stretches on two-filament bundles compared with the intensity of cofilin clusters on single actin filaments (N = 3 experiments, with n = 23 (blue), 8 (green), 8 (orange) spots analyzed on two-filament bundles, and n = 35, 5, 18 cofilin clusters on single filaments, respectively). Large symbols represent averages over individual measures from independent experiments.

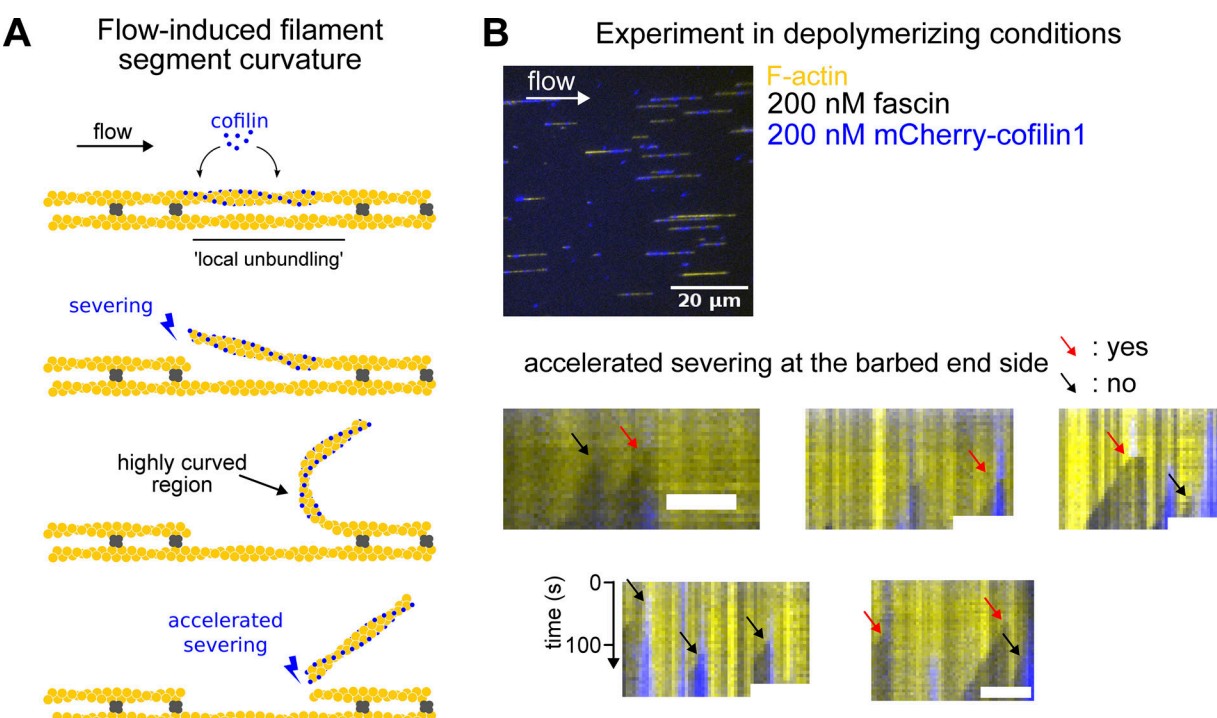

Figure S8.   **Curvature of cofilin-actin segments induced by the flow may accelerate severing. (A)** Schematics illustrating how a microfluidics flow may accelerate severing at the barbed end side of a cofilin-decorated segment. Once a cofilin cluster has severed at its pointed end side, the flow may induce a high curvature at its barbed end side, which results in an increase of the severing at this boundary (Wioland et al., 2019b). **(B)** Microfluidics assay where single filaments and two-filament bundles are exposed to fascin and cofilin in the absence of actin. Severing at the pointed end side of cofilin clusters creates free barbed ends which depolymerize. Representative kymographs are shown (scale bar = 5 μm). Red arrows indicate pointed end side severing events which are accompanied with the departure of the cofilin cluster (i.e., due to accelerated severing at the barbed end side, as depicted in A), and black arrows severing events where it is not. 46% of the severing events leads to accelerated cofilin-segment departure ($n$ = 26 events).

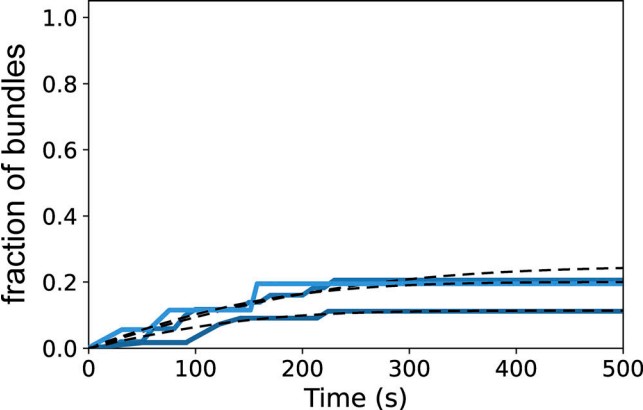

Figure S9.   **Fraction of cofilin clusters that will sever before a cofilin cluster is initiated on an adjacent filament of a two-filament bundle in the region facing the first cofilin cluster.** Fraction of cofilin clusters that will sever before a cofilin cluster is initiated on an adjacent filament of a two-filament bundle in the region facing the first cofilin cluster, as a function of time, from three independent experiments ($n$ = 101, 17, and 28 cofilin clusters, from top to bottom curves). Fits of the curves (dashed lines, see Materials and methods) yield fractions of 0.3, 0.2, and 0.15.

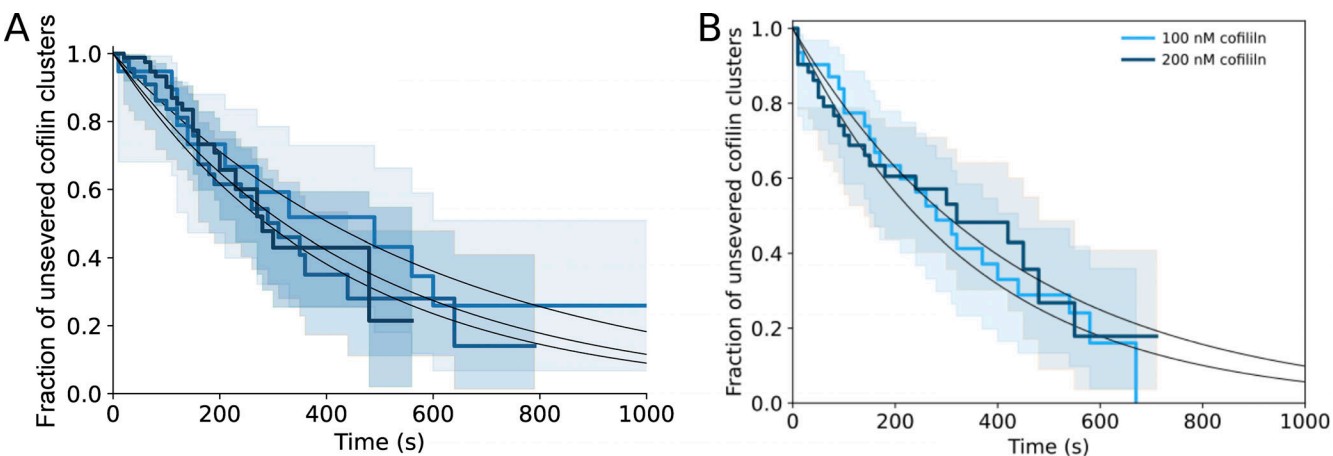

Figure S10. **Cofilin cluster severing rate on single actin filaments. (A)** Cofilin cluster severing rate on single actin filaments from three independent experiments ($n$ = 19, 88, 45 filaments from lighter to darker blue curves). Single exponential fits (black lines) yield cofilin cluster severing rates on single filaments of 1.7, 2.15, and 2.4 × $10^{-3}$ s$^{-1}$. **(B)** Cofilin cluster severing rate on single actin filaments at 100 and 200 nM cofilin ($n$ = 31 and 53 filaments, respectively). Single exponential fits (black lines) yield cofilin cluster severing rates of 2.86 and 2.31 × $10^{-3}$ s$^{-1}$.

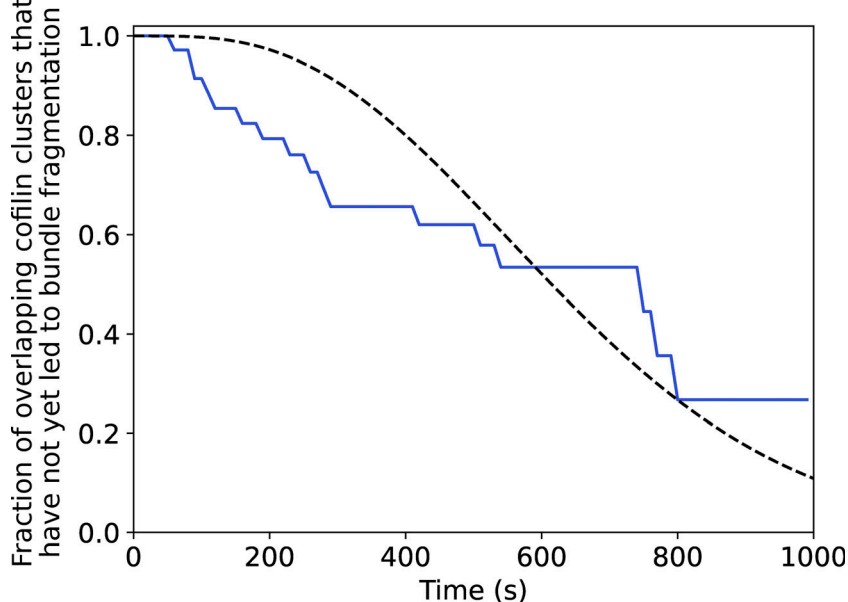

Figure S11. **Fraction of colocalized cofilin clusters leading to two-filament bundle fragmentation.** Related to Fig. 5. At 100 nM cofilin, a fraction of colocalized cluster regions on two-filament bundles that have not yet led to the fragmentation of the bundle, versus time ($n$ = 36 events). The dashed line shows the best fit by a computed curve (with $k_{init,2}$ = 8 × $k_{init,SF}$, see Materials and methods), using a chi-square minimization procedure, yielding a cofilin cluster severing rate $k_{sev,2}$ = 2.7 × $10^{-3}$ s$^{-1}$ (±0.8, SD), close to the one observed on single filaments (see Fig. S10).

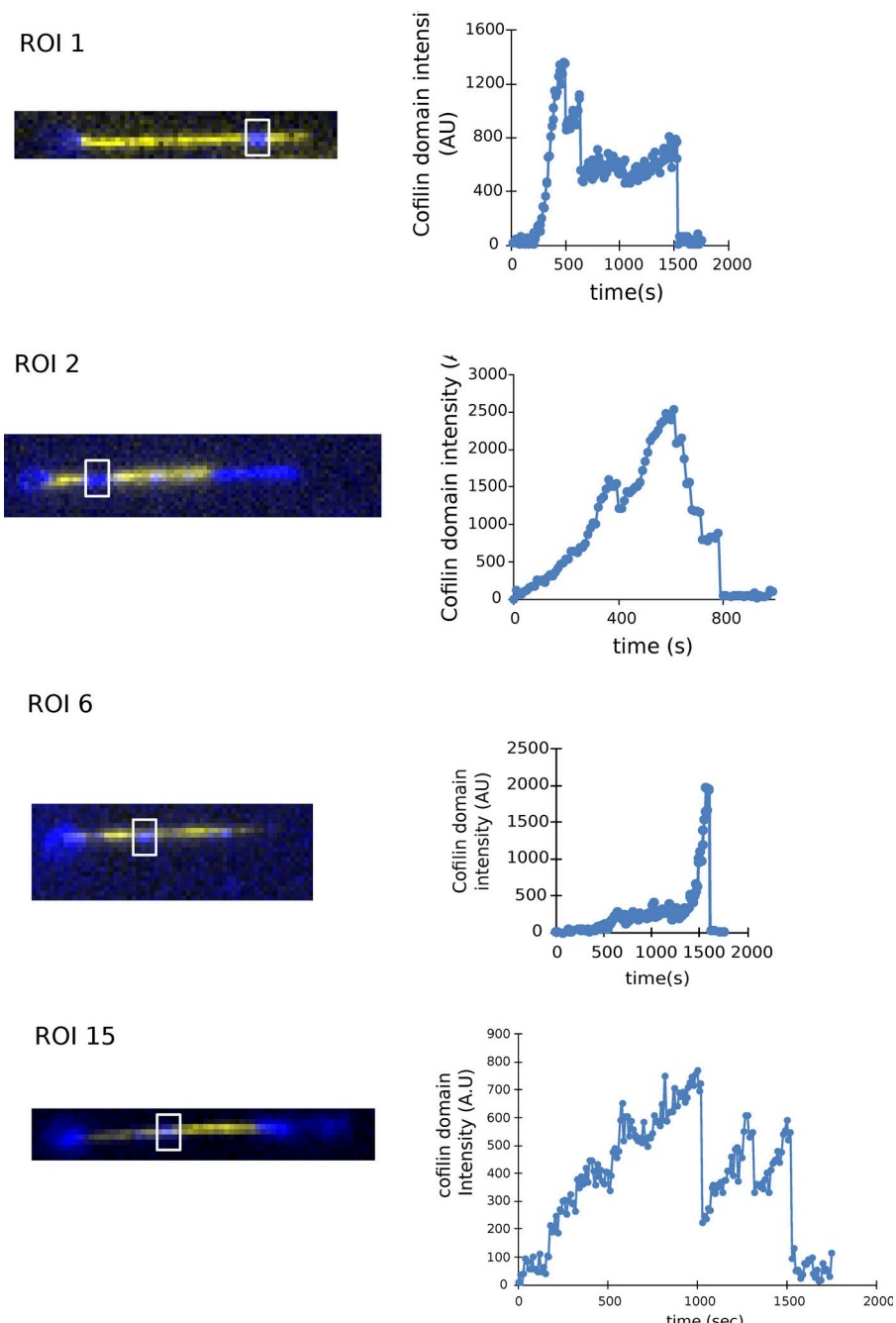

Figure S12.   **Observations of colocalized cofilin clusters on bundles composed of more than two filaments.** Cofilin fluorescence intensity of cofilin clusters on bundles larger than two filaments, showing several colocalized cofilin clusters, with multiple decreasing steps indicating cluster severing.

**200 nM fascin**

**200 nM fascin + 500 nM cofilin**

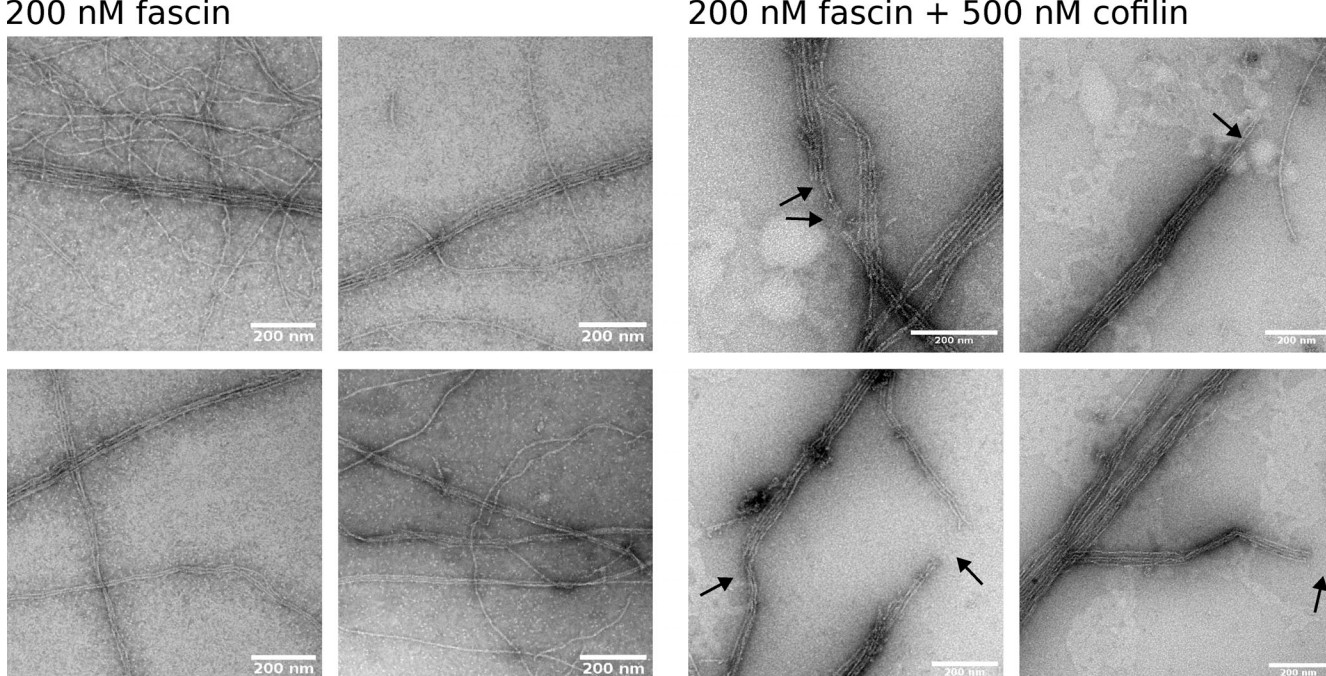

**Figure S13.   Negative staining EM images of fascin-induced actin filaments exposed to cofilin.** Preformed 1 µM F-actin filament bundles, 10× diluted in a F-buffer solution containing 200 nM fascin, (top) with or (bottom) without 500 nM cofilin, were deposited on an EM grid and negatively stained. Bundles display clear cuts in the presence of cofilin (black arrows) with filament ends extending over 60 nm (±42 nm, SD, n = 37 breaks analyzed). Scale bars are 200 nm.

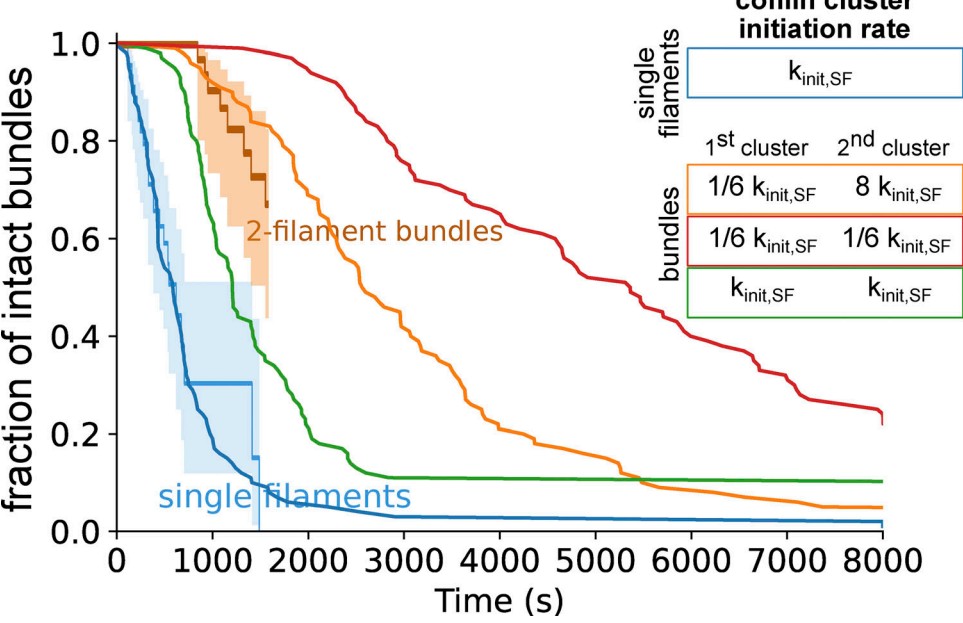

**Figure S14.   Numerical simulations indicating the impact of inter-filament cooperativity for the initiation of cofilin clusters on the fragmentation of two-filament bundles.** Fraction of intact single filaments or two-filament bundles over time (n = 100 simulated two-filament bundles for each condition) upon exposure to cofilin. The reference initiation rate of cofilin clusters is the one on single filaments (blue curve, $k_{init,SF}$). On bundles, the initiation rate of the first cofilin cluster is either the one observed experimentally for fascin-induced two-filament bundles (orange and red curves, $1/6\ k_{init,SF}$) or similar to the one measured on single actin filaments (green curve, $k_{init,SF}$). The initiation rate of a second cofilin cluster, i.e., on a filament in the region facing a first cofilin cluster on the other filament, either results from interfilament cooperativity imposed by fascin bundling (see main text) (orange curve, $8 × k_{init,SF}$) or not (red curve, $1/6\ k_{init,SF}$, and green curve, $k_{init,SF}$). Experimental data for single actin filaments and two-filament bundles (from Fig. 2 B) are shown for comparison.

**Figure S15.  Twist-constraining two-filament bundles lead to its faster fragmentation by cofilin.** Survival fractions of unsevered 5-µm long segments from unanchored bundles (twist-unconstrained, *n* = 20) or anchored bundles (twist-constrained, *n* = 16) as a function of time, when exposed to 200 nM mCherry-cofilin1 and 200 nM fascin. 95% confidence intervals are shown as shaded surfaces. Log-rank test P value <0.005.

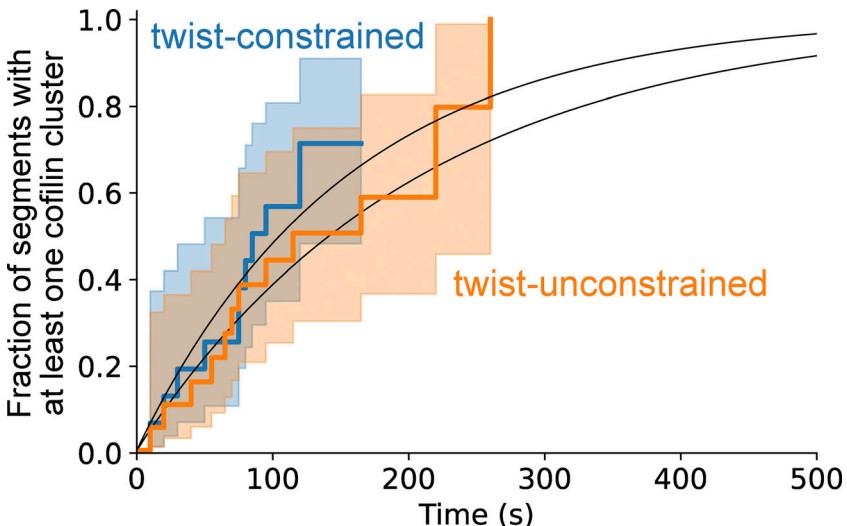

**Figure S16.  Initiation of cofilin clusters on twist-constrained filaments.** Fraction of 5-µm segments with at least one cofilin cluster on twist-constrained (*n* = 16 segments, blue) or twist-unconstrained (*n* = 20 segments, orange) bundles. Log-rank test P value = 0.45.

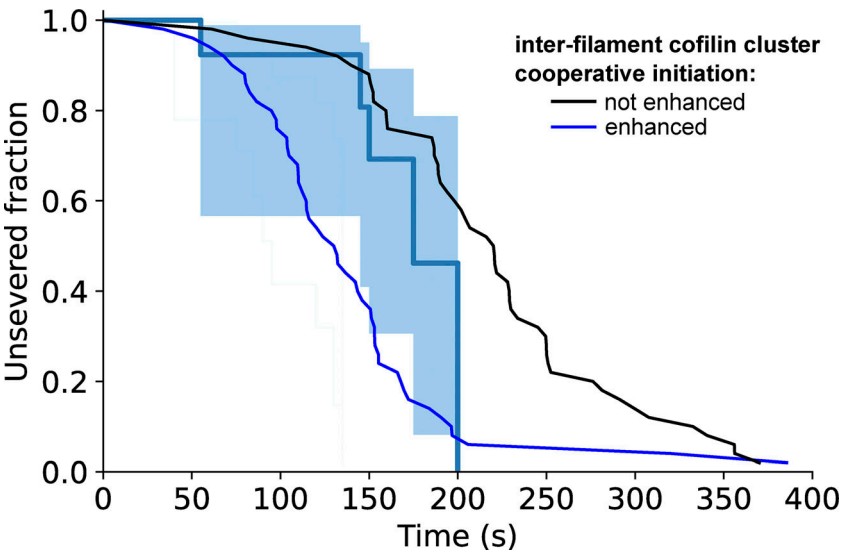

Figure S17.   **Numerical simulations showing the impact of inter-filament cooperativity for the initiation of cofilin clusters on the fragmentation of twist-constrained two-filament bundles.** Results of numerical simulations show that fragmentation of twist-constrained two-filament bundles (*n* = 50 simulated two-filament bundles) is faster with interfilament cooperativity for the initiation of cofilin clusters than without. Simulations with no interfilament cooperativity seem to better reflect the experimental observations (data from Fig. 6 C, *n* = 16 segments of two-filament bundles).

Video 1.   **Fascin-induced actin filament bundles exposed to cofilin, in an "open chamber" assay.** In an open chamber, Alexa488-actin filaments are grown and aged for 15 min from surface-anchored spectrin seeds in the presence of 200 nM human-fascin1, in a buffer containing 0.2% methylcellulose. They are subsequently exposed to 80 nM mCherry-cofilin1, 0.15 μM 10% Alexa488-actin, and 200 nM fascin. Images are acquired using TIRF microscopy, 1 image every 10 s. Fluorescent actin appears in yellow, fluorescent cofilin in blue. Scale bar: 20 μm. Image display rate: 9 frames/s. This video is related to Fig. 1.

Video 2.   **Single actin filaments exposed to cofilin, in an "open chamber" assay.** In an open chamber, Alexa488-actin filaments are grown and aged for 15 min from surface-anchored spectrin seeds without human-fascin1 in a buffer containing 0.2% methylcellulose (not shown). The movie starts when they get subsequently exposed to 80 nM mCherry-cofilin1, 0.15 μM 10% Alexa488-actin, and 200 nM fascin. Images were acquired using TIRF microscopy, 1 image every 10 s. Fluorescent actin appears in yellow, fluorescent cofilin in blue. Scale bar: 20 μm. Image display rate: 9 frames/s. This video is related to Fig. 1.

Video 3.   **Fascin-induced actin filament bundles exposed to cofilin, in a microfluidics assay.** In a microfluidics chamber, Alexa488-actin filaments are grown from surface-anchored spectrin seeds before being aged for 15 min while being exposed to 200 nM human-fascin1 to induce bundling. The movie starts when they get subsequently exposed to 200 nM mCherry-cofilin1, 0.15 μM 10% Alexa488-actin, and 200 nM fascin. Images are acquired using epifluorescence microscopy, 1 image every 10 s. Fluorescent actin appears in yellow, fluorescent cofilin in blue. Scale bar: 20 μm. Image display rate: 9 frames/s. This video is related to Fig. 2.

**Provided online are Table S1 and Data S1. Table S1 shows cofilin reaction rates on single and bundle actin filaments. Data S1 shows the data for all the plots shown in the figures of the article. Each figure panel corresponds to a different sheet.**

