## [Peer Review File · The Journal of Cell Biology]

Fascin-induced bundling protects actin filaments from disassembly by cofilin

Jahnvi Chikireddy, Leana Lengagne, Remi Le Borgne, Catherine Durieu, Hugo Wioland, Guillaume Romet-Lemonne, and Antoine Jégou

Corresponding Author(s): Antoine Jégou, Institut Jacques Monod

Review Timeline:

Submission Date:	2023-12-20
Editorial Decision:	2024-02-02
Revision Received:	2024-02-26

Monitoring Editor: Pekka Lappalainen

Scientific Editor: Dan Simon

Transaction Report:

DOI: <https://doi.org/10.1083/jcb.202312106>

Revision 0

Review #1

1. Evidence, reproducibility and clarity:

Evidence, reproducibility and clarity (Required)

In this study, it is shown that cofilin severs actin filaments slowly when fascin is present. Authors show that this is due to slower cluster nucleation of cofilin on fascin-induced actin bundles. Interestingly, the authors show that cofilin binding promotes helicity in actin filament bundles which in turn promotes fascin exclusion and more cofilin clustering in adjacent filament bundles; thus, inducing local transmission of structural changes.

The authors use an elegant approach, and the data is nicely presented. Overall, I consider that this manuscript is in good shape to be published. It might benefit from language editing, though.

2. Significance:

Significance (Required)

According to me the significance of this manuscript is that elegantly shows the molecular details of the cofilin severing effect of fascin-induced actin filament bundles. The authors show that cofilin binding promotes helicity in actin filament bundles which in turn promotes fascin exclusion and more cofilin clustering in adjacent filament bundles; thus, inducing local transmission of structural changes.

3. How much time do you estimate the authors will need to complete the suggested revisions:

Estimated time to Complete Revisions (Required)

(Decision Recommendation)

Cannot tell / Not applicable

4. *Review Commons* values the work of reviewers and encourages them to get credit for their work. Select 'Yes' below to register your reviewing activity at Web of Science

Reviewer Recognition Service (formerly Publons); note that the content of your review will not be visible on Web of Science.

Yes

Review #2

1. Evidence, reproducibility and clarity:

Evidence, reproducibility and clarity (Required)

****Summary:****

In this study, Chikireddy et al. perform a series of experiments in which they compare the efficiency of cofilin-mediated severing and actin filament disassembly on individual filaments versus bundles of different sizes from by the actin-bundling protein fascin. The key outcome, quite distinct from previously published conclusions by the authors themselves and other authors, is that fascin bundling actually reduces cofilin-mediated severing mostly because of much slower "nucleation" of cofilin clusters on fascin-bound filament bundles. Cofilin cluster formation is followed by local fascin removal, and the nucleation of a cofilin cluster on an adjacent bundle in the absence of fascin is strongly enhanced. The reason for the latter surprising observation is not entirely clear, but proposed to arise from cofilin-mediated changes in filament helicity of neighboring filaments. To my understanding, the main reason why fascin protects from cofilin severing here rather than enhancing it (as reported previously) is due to the lack of constraining of the induced, cofilin-mediated twist, because if this twist is constrained e.g. by anchoring of the bundles to the surface chamber, then severing by cofilin is accelerated.

****Major comments:****

I think the study is very well done, most experiments are super-elegant and controlled; I really don't have any objections against the conclusions drawn, as most of what I have seen is totally justified and reasonable. So from a scientific point of view, I can easily agree with all the major conclusions drawn, and so in my view, this should be published fast.

****Minor comments:****

There are two minor points that could be addressed:

1. I am not entirely convinced by the conclusions drawn from the EM images shown in Figure

6A, and in particular by the filaments in two-filament bundles locally twisting around each other (without breaking) at spatial sites lacking fascin and decorated by cofilin. This is hard to imagine for me, and the evidence for something like this happening is not very strong, as in the EM, only larger bundles could be observed. In addition, I am not sure that the braiding of filaments seen in the presence of cofilin is really occurring just locally on cofilin-decorated bundle segments and thus indeed coincides with loss of fascin as proposed in the scheme in Fig. 6B.

Can the authors exclude that the braiding is not caused by some experimental artefact, as induced perhaps by sample preparation for negative staining? Did the authors quantify the occurrence of such braided bundle segments with and without cofilin? How large are these braided segments on average when you quantify them? Would you also see them if you prepared the bundles for an alternative EM-technique, such as Cryo-EM, for instance? This may admittedly all be experimentally challenging, but would it be possible to combine the negative staining of filaments with staining for cofilin and/or fascin using immunogold technology, to prove that the braided segments do indeed correlate with high cofilin and low fascin concentrations? In the absence of such data, and in particular in the absence of a clear quantification, the proposal is too strong in my view. Finally, it would be nice (albeit not essential I guess) to also look at two-filament bundles. The authors stated these can not be easily generated due to the tendency of fascin to promote the formation of larger bundles, but can this not be titrated/tuned somehow by lowering fascin concentrations, to come closer in reality to what is proposed to occur in the scheme in Figure 6B? In any case, the way the data are presented right now appears to constitute a pretty large gap between experimental evidence and theoretical model.

2. I think that the proposal of cofilin-decorated filaments to "transfer" the resulting cofilin-induced changes in filament helicity onto neighboring filaments in the bundle, which is proposed to occur locally and in the absence of fascin is a bit vague, and difficult to understand mechanistically. Can the authors speculate, at least, how they think this would occur? Are there no alternative possibilities for explaining obtained results? Maybe I am missing something here, but with considering cofilin to be monomeric and only harboring one actin-binding site, this proposal of helicity transfer onto neighboring filaments seems inconclusive.

2. Significance:

Significance (Required)

General assessment:

The strength of this study is that owing, at least in part, to the microfluidics devices employed and the careful biochemistry, the experimental setups are super-controlled and clean, and they are used in a highly innovative and elegant fashion. The simulations are also nice! A limitation is that it is not entirely clear how precisely the main observations can be translated to what's happening in vivo. The results are largely dependent on the bundles not being constrained I understand, so to what extent would bundles be unconstrained in vivo? Perhaps this is not so important, because the experimental setup allows the authors to dissect specific biochemical behaviors and inter-dependencies between distinct actin binding proteins, but the latter view (if correct) could be stated more clearly!

Advance:

As stated above, the results are opposite to the proposed synergistic activities of fascin and cofilin observed for bundles previously, perhaps because they were not constrained. So although touched in part and in a very polite fashion in the discussion, the authors could specify more clearly what the differences between the studies are, and which of the distinct activities observed either here or in previous literature will be dominant or more relevant to consider in the future? This will be hard to discern as is now, in particular for non-experts.

Audience:

This manuscript will be most influential for a specialized audience interested in the complexities of biochemical activities of specific actin binding proteins when looking at them in combination. Although specialized, this is still a quite relevant audience though, since prominent actin binding proteins like cofilin are highly important in virtually any cell type and various actin structures, hence of broad relevance again in this respect.

Expertise:

I am a cell biologist and geneticist interested in actin dynamics and actin-based, motile processes.

3. How much time do you estimate the authors will need to complete the suggested revisions:

Estimated time to Complete Revisions (Required)

(Decision Recommendation)

Between 3 and 6 months

Yes

Review #3

1. Evidence, reproducibility and clarity:

Evidence, reproducibility and clarity (Required)

My only major concern is that although the authors provide data that strongly supports interfilament cooperativity in two filament bundles for cofilin binding, the evidence to support that this induces filament twist on the opposing filament is not strong enough to conclusively establish this as the mechanism for the observed interfilament cooperativity. This is stated as such in the results section as a proposed model, but stated with more certainty than the presented data supports in the discussion. It might be better, based on the data presented, to state this as one possible mechanism for the observed cooperativity.

Areas within the paper, if addressed, will improve the arguments presented as well as the readability of the paper.

1. The authors use both the terms cofilin binding (in section I of the results) as well as cofilin nucleation (in section III of the results). It is unclear if these terms are meant to indicate the same, or different, processes. The manuscript would benefit from a clear explanation of the steps of cofilin-mediated disassembly measured and quantified in the experiments, namely nucleation (or binding), cluster growth, and filament or bundle fragmentation. A clear description of these steps would also allow the reader to follow the logic of the experiments from Figure 3 to Figure 5.
2. Throughout the paper, the authors move from single filaments, to 2-filament bundles, to multifilament bundles, using different concentrations of fascin and cofilin. Given the biphasic behavior of cofilin, namely that low concentrations favor severing and high concentrations can favor coating and filament stabilization, I think it is important that concentrations for the components are consistent across experiments, and if changes of concentrations of important components (such as cofilin and fascin) are changed, a clear explanation as to why is included.
3. In Figure 2, it is mentioned that for the spectrin seeds with the microfluidics, the filaments consisting of larger bundles were not analyzed along with the single filament and 2-filament bundles. Instead, a different experiment with seeds attached to beads is used to assess larger filament bundles. Why were larger bundles not analyzed in the microfluidic experiment? And conversely, why were 2 filament bundles not assessed with the beads? Comparing the findings on two filament bundles with the findings on multifilament bundles would be easier for the reader if the small and large bundles were evaluated in the same experiments. If this is not experimentally feasible, the authors need to provide clearer explanation as to why this analysis is not included.
4. The authors indicate that at increased fascin concentration (1 μ M) that single filaments decrease the nucleation rate of cofilin clusters. The authors should comment on the mechanism for fascin (at 1 μ M concentration) for affecting cofilin binding.
5. The authors should determine and include the dissociation rate for the labeled cofilin used in this study, especially given the proposed mechanism for cofilin excluding fascin within the bundles.

6. For Figure 4, D and E, what do the dynamics of fascin and cofilin signal look like on a larger filament bundle? It would be informative to provide the cofilin cluster nucleation rate on larger filament bundles with a range of fascin concentrations (as in 3D for a two filament bundle).
7. Additionally, it would be useful to report the cofilin severing rate at a range of cofilin concentrations, at least for the 2 filament bundles. When the severing occurs in the two filament bundles, does this severing occur mostly at boundaries with cofilin-actin and bare actin or does this severing occur at cofilin-actin/fascin-actin boundaries?
8. For the images of large bundles appearing braided in figure 6A, the lower left panel the braided appearance is not obvious. Additionally, what is the number of filaments in the bundles shown? Finally, given that in Figure 3F it is indicated that cofilin cluster nucleation events are rare on large bundles, and the cluster growth rate is reduced on large bundles (Figure 4C), the authors need to indicate how frequently this braided appearance is observed as well as what the nucleation rate, growth rate and severing rate is for 500nM cofilin on bundles.
9. The authors indicate that the rapid fragmentation of twist constrained 2-filament bundles prevented them from directly quantifying the nucleation rate of the subsequent cofilin clusters that overlapped the initial ones. I'm unclear why this is the case, and if this is the case, I don't understand how the authors can be sure that a second nucleation event occurred in the twist constrained bundles. From the experimental data in 7C, it appears that the fragmentation rate for two filament bundles is similar to the fragmentation rate for twist constrained single filaments. The authors need to clearly state what they were able to observe and quantify as well as include the timing for this severing. If the authors could not observe a second nucleation event prior to severing, this should be clearly stated. This could be due to the rapid fragmentation, but it could also be due to severing occurring in the absence of a second cofilin nucleation event. It would be informative to compare the time from cofilin nucleation to severing event for two filament bundles in twist constrained and unconstrained. Clarification of the dynamics of nucleation and spreading of cofilin and the timing of fragmentation of the twist constrained filament bundles is needed.
10. Discussion of how twist constrained fragmentation dynamics might affect the dynamics of larger bundles in structures such as filopodia would be useful.

****Minor changes that would improve the paper:****

11. In Figure 1C, Figure 2B and Figure 2E, the indication, on the graph, of the fold-change between the rates is confusing as it is not clear from the labeling on the graph that the x15 is referring to the slope of the lines, keeping this information in the legend is appropriate, but if it is to be included on the graph, perhaps adding in the linear fit on the graph is also needed.
12. Figure 7A, lining up the diagram with the kymographs below would help improve interpretation of the diagram and simulation. Alternatively, if the diagram (upper) in A does not diagram the kymographs below, this needs to be clearly stated, and it would be preferable that the diagram above matches the kymographs below.
13. Despite referencing the Breitsprecher, 2011 paper in the introduction, the authors do not explain how their results showing that cofilin fragments filament bundles slower than single actin filaments correspond with the Breitsprecher findings that fascin bundles favors cofilin filament severing. While the authors do not need to explain the Breitsprecher data, if they reference these findings that run counter to their results, an explanation for the discrepancy would be reasonable to include in the discussion.

2. Significance:

Significance (Required)

The research presented in "Fascin-induced bundling protects actin filament from disassembly by cofilin" is relevant and of interest to the field as it directly addresses a limitation in our understanding of how cofilin-induced severing occurs in F-actin bundles. Bundled F-actin may constitute the majority of linear F-actin within the cell and is specifically important in F-actin-based structures such as filopodia and stress-fibers. The data supports a model for interfilament cooperativity that provides a molecular mechanism for cofilin-mediated severing of fascin-bundled filaments.

3. How much time do you estimate the authors will need to complete the suggested revisions:

Estimated time to Complete Revisions (Required)

(Decision Recommendation)

Less than 1 month

No

Manuscript number: RC-2023-02015

Corresponding author(s): Antoine, Jégou

1. General Statements

Dear editor,

We are submitting a fully revised version of our manuscript in light of the points raised by the reviewers that we hope to have addressed convincingly below.

As you will see, the 3 reviewers are positive about the impact of our manuscript:

- Reviewer #1 states that **“The authors use an elegant approach, and the data is nicely presented. Overall, I consider that this manuscript is in good shape to be published.”**
- Reviewer #2 highlights the **“broad relevance [...]”** of this work, and writes **“I can easily agree with all the major conclusions drawn, and so in my view, this should be published fast”**. Reviewer #2 also highlights that **“the experimental setups are super-controlled and clean, and they are used in a highly innovative and elegant fashion. The simulations are also nice!”**.
- For reviewer #3, this paper is **“of interest to the field as it directly addresses a limitation in our understanding of how cofilin-induced severing occurs in F-actin bundles.”**

The reviewers' main concern was that, while they agreed that there is interfilament cooperativity for the nucleation of cofilin clusters, they were less convinced by the super-twisting mechanism that we propose to explain it. In particular, they found that our EM data did not support this mechanism. We agree that this point is not essential, and we have moved our EM data to the supplementary section. As suggested by reviewer #3, we now make a clearer distinction between the interfilament cooperativity that we discovered, and the super-twisting mechanism we propose as a possible interpretation.

Furthermore, as requested by reviewer #2 and #3, we have substantially improved our discussion, by discussing in more detail the implications of our results in the cell context, in particular for filopodia.

We believe that our manuscript, now in its revised version, should alleviate the reviewers' concerns. We expect it should represent a significant advance in our understanding of how the architecture of the actin cytoskeleton is key in regulating its disassembly by ADF/cofilin.

Reviewer #1 (Evidence, reproducibility and clarity (Required)):

In this study, it is shown that cofilin severs actin filaments slowly when fascin is present. Authors show that this is due to slower cluster nucleation of cofilin on fascin-induced actin bundles. Interestingly, the authors show that cofilin binding promotes helicity in actin filament bundles which in turn promotes fascin exclusion and more cofilin clustering in adjacent filament bundles; thus, inducing local transmission of structural changes.

The authors use an elegant approach, and the data is nicely presented. Overall, I consider that this manuscript is in good shape to be published. It might benefit from language editing, though.

We thank the reviewer for their positive comments. We have edited the manuscript to improve its readability (changes are in blue in the manuscript).

Reviewer #1 (Significance (Required)):

According to me the significance of this manuscript is that elegantly shows the molecular details of the cofilin severing effect of fascin-induced actin filament bundles. The authors show that cofilin binding promotes helicity in actin filament bundles which in turn promotes fascin exclusion and more cofilin clustering in adjacent filament bundles; thus, inducing local transmission of structural changes.

Reviewer #2 (Evidence, reproducibility and clarity (Required)):

Summary:

In this study, Chikireddy et al. perform a series of experiments in which they compare the efficiency of cofilin-mediated severing and actin filament disassembly on individual filaments versus bundles of different sizes from by the actin-bundling protein fascin. The key outcome, quite distinct from previously published conclusions by the authors themselves and other authors, is that fascin bundling actually reduces cofilin-mediated severing mostly because of much slower "nucleation" of cofilin clusters on fascin-bound filament bundles. Cofilin cluster formation is followed by local fascin removal, and the nucleation of a cofilin cluster on an adjacent bundle in the absence of fascin is strongly enhanced. The reason for the latter surprising observation is not entirely clear, but proposed to arise from cofilin-mediated changes in filament helicity of neighboring filaments. To my understanding, the main reason why fascin protects from cofilin severing here rather than enhancing it (as reported previously) is due to the lack of constraining of the induced, cofilin-mediated twist, because if this twist is constrained e.g. by anchoring of the bundles to the surface chamber, then severing by cofilin is accelerated. We thank the reviewer for their positive feedback on the manuscript. We have substantially edited the manuscript in light of the insightful comments of the reviewer (changes are in blue).

Major comments:

I think the study is very well done, most experiments are super-elegant and controlled; I really don't have any objections against the conclusions drawn, as most of what I have seen is totally justified and reasonable. So from a scientific point of view, I can easily agree with all the major conclusions drawn, and so in my view, this should be published fast.

Minor comments:

There are two minor points that could be addressed:

1) I am not entirely convinced by the conclusions drawn from the EM images shown in Figure 6A, and in particular by the filaments in two-filament bundles locally twisting around each other (without breaking) at spatial sites lacking fascin and decorated by cofilin. This is hard to imagine for me, and the evidence for something like this happening is not very strong, as in the EM, only larger bundles could be observed. In addition, I am not sure that the braiding of filaments seen in the presence of cofilin is really occurring just locally on cofilin-decorated bundle segments and thus indeed coincides with loss of fascin as proposed in the scheme in Fig. 6B.

Can the authors exclude that the braiding is not caused by some experimental artefact, as induced perhaps by sample preparation for negative staining?

We thank the reviewer for raising this point. We have repeated the negative staining EM experiments several times and now show new images and quantification (new Supp Fig. 13). In our new series of experiments, the braiding that was previously shown in Fig. 6 proved difficult to reproduce and to quantify. We therefore decided to remove EM observations from the main Fig. 6, and we no longer present them as evidence supporting the mechanism that we propose for inter-filament cooperativity.

From EM images, we now quantify the frequency of fragmentation of large actin filament bundles. We observed that bundles often terminate with the ends of their filaments in close proximity, consistent with sharp breaks due to co-localized cofilin clusters.

We have rewritten this part of the result section in the manuscript which now reads : 'To further investigate larger bundles, we imaged them using negative staining electron microscopy. In the absence of cofilin, filaments in bundles are arranged in a parallel manner, as previously reported in vitro (Jansen et al, 2011). Compared with the control, filament bundles exposed to cofilin show numerous sharp breaks (65 breaks per 122 μm of bundles, versus 4 breaks per 68 μm in the control. Supp. Fig. 13). This is consistent with bundle fragmentation occurring at boundaries of co-localized cofilin clusters.'

Did the authors quantify the occurrence of such braided bundle segments with and without cofilin?

How large are these braided segments on average when you quantify them? Would you also see them if you prepared the bundles for an alternative EM-technique, such as Cryo-EM, for instance?

As mentioned in the answer to the previous point, the braided segments proved difficult to reproduce and quantify, and we have removed EM experiments from the main figure 6. Instead of the braided segments, we now quantify the severing of the bundles, and the distribution of filament ends at the extremities of the bundles (new Supp. Fig. 13).

We have not tried Cryo-EM due to limited access to such experimental tools within the timeframe of the study.

This may admittedly all be experimentally challenging, but would it be possible to combine the negative staining of filaments with staining for cofilin and/or fascin using immunogold technology, to prove that the braided segments do indeed correlate with high cofilin and low fascin concentrations? In the absence of such data, and in particular in the absence of a clear quantification, the proposal is too strong in my view. Finally, it would be nice (albeit not essential I guess) to also look at two-filament bundles. The authors stated these can not be easily generated due to the tendency of fascin to promote the formation of larger bundles, but can this not be titrated/tuned somehow by lowering fascin concentrations, to come closer in reality to what is proposed to occur in the scheme in Figure 6B? In any case, the way the data are presented right now appears to constitute a pretty large gap between experimental evidence and theoretical model.

We agree with the reviewer that EM observations are limited and, alone, do not provide strong evidence in favor of braiding/super-twisting being the mechanism responsible for inter-filament cooperativity (please see our answers to the points above). We have performed negative staining EM assays at higher cofilin-1 concentration (500 nM) compared to microfluidics assays, in order for cofilin to quickly bind to filaments, even in large bundles, so that our chances to capture bundles targeted by cofilin would be high.

Nevertheless, both microfluidics and EM observations point in the same direction : bundle fragmentation by cofilin is caused by the co-localized cooperative nucleation of cofilin clusters.

2) I think that the proposal of cofilin-decorated filaments to "transfer" the resulting cofilin-induced changes in filament helicity onto neighboring filaments in the bundle, which is proposed to occur locally and in the absence of fascin is a bit vague, and difficult to understand mechanistically. Can the authors speculate, at least, how they think this would occur? Are there no alternative possibilities for explaining obtained results? Maybe I am missing something here, but with considering cofilin to be monomeric and only harboring one actin-binding site, this proposal of helicity transfer onto neighboring filaments seems inconclusive.

On single actin filaments, the change of helicity induced by cofilin binding has been observed by many groups using EM and cryoEM (e.g. McGough et al, JBC 1997 10.1083/jcb.138.4.771; Egelman et al, PNAS 2011 10.1073/pnas.1110109108 ; Huehn et al, JBC 2018 10.1074/jbc.AC118.001843). These studies have revealed that actin subunits get 'tilted' relative to their original orientation along the filament long axis. This leads to the shortening of the helical pitch for cofilin-saturated actin filament segments.

In our assays, the progressive binding of cofilin along a single filament creates a cluster where all actin subunits are tilted and the helical pitch of the filaments within the cluster is shortened (from a half pitch of 36 nm down to 27 nm). This change of helicity in a cluster induces the rotation of one end of the filament relative to the other (as we have shown previously in Wioland et al, PNAS 2019). Therefore, if two parallel filaments are stapled together, the local twisting of one filament causes the twisting of the other in the overlapping region.

We have rephrased this point to more clearly explain this in the last paragraph of the results section:

"From our kinetic analysis, we propose the following model that recapitulates the binding of cofilin to fascin-induced 2-filament bundles (Fig. 6D). Initially, actin filaments in fascin-induced bundles are in conformations that are less favorable for cofilin binding than isolated actin filaments. Once a cofilin cluster has nucleated, its expansion locally triggers fascin unbinding and prevents it from rebinding. The increase of filament helicity induced by cofilin causes a local twisting of the entire bundle, thereby changing the helicity of the adjacent filament in the fascin-free region facing the cofilin cluster. In this region, the increase in filament helicity enhances cofilin affinity, and thus locally promotes the nucleation of a cofilin cluster (inter-filament cooperativity)."

We have tried to think of other alternative scenarios that might explain our observations, but none appeared to be valid.

Reviewer #2 (Significance (Required)):

General assessment:

The strength of this study is that owing, at least in part, to the microfluidics devices employed and the careful biochemistry, the experimental setups are super-controlled and clean, and they are used in a highly innovative and elegant fashion. The simulations are also nice! A limitation is that it is not entirely clear how precisely the main observations can be translated to what's

happening in vivo. The results are largely dependent on the bundles not being constrained I understand, so to what extent would bundles be unconstrained in vivo? Perhaps this is not so important, because the experimental setup allows the authors to dissect specific biochemical behaviors and inter-dependencies between distinct actin binding proteins, but the latter view (if correct) could be stated more clearly!

We thank the reviewer for their remarks. We have updated the part where we discuss the biological implications of our in vitro observations to better explain how the twist-constraints expected for fascin bundles in cells would accelerate cofilin bundle disassembly.

Advance:

As stated above, the results are opposite to the proposed synergistic activities of fascin and cofilin observed for bundles previously, perhaps because they were not constrained. So although touched in part and in a very polite fashion in the discussion, the authors could specify more clearly what the differences between the studies are, and which of the distinct activities observed either here or in previous literature will be dominant or more relevant to consider in the future? This will be hard to discern as is now, in particular for non-experts.

We agree with the reviewer that the manuscript will benefit from discussing more in depth the plausible reasons why our experimental observations are in disagreement with the earlier interpretation by Breitsprecher and colleagues. We have extended our discussion on this point, which now reads: “Previously, using pyrene-actin bulk experiments, Breitsprecher and colleagues observed a diminished cofilin binding to fascin-induced filament bundles (Breitsprecher et al, 2011). In spite of this, their observation of fluorescently labeled actin filament bundles seemed to indicate an efficient severing activity. Since cofilin was not fluorescently labeled, they could not observe cofilin clusters, and they proposed that severing was enhanced because fascin served as anchors along filaments and impeded cofilin-induced changes in filament helicity”

Audience:

This manuscript will be most influential for a specialized audience interested in the complexities of biochemical activities of specific actin binding proteins when looking at them in combination. Although specialized, this is still a quite relevant audience though, since prominent actin binding proteins like cofilin are highly important in virtually any cell type and various actin structures, hence of broad relevance again in this respect.

Expertise:

I am a cell biologist and geneticist interested in actin dynamics and actin-based, motile processes.

Reviewer #3 (Evidence, reproducibility and clarity (Required)):

My only major concern that is that although the authors provide data that strongly supports interfilament cooperativity in two filament bundles for cofilin binding, the evidence to support that this induces filament twist on the opposing filament is not strong enough to conclusively establish this as the mechanism for the observed interfilament cooperativity. This is stated as such in the results section as a proposed model, but stated with more certainty than the presented data supports in the discussion. It might be better, based on the data presented, to state this as one possible mechanism for the observed cooperativity.

We thank the reviewer for their remark. We have edited our discussion section to clearly say that inter-filament cooperativity arises from cofilin-induced filament twisting is a proposed model that would best account for what we observed: “Indeed, we report here the exclusion of fascin from within cofilin clusters, and a strong increase in the nucleation of cofilin clusters on adjacent filaments. This inter-filament cooperativity mechanism leads to the co-localized nucleation of cofilin clusters, and permits bundle fragmentation faster than if the nucleation of cofilin clusters on adjacent filaments were purely random. To our knowledge, this is the first time such inter-filament cooperativity is ever reported. To explain this mechanism, we propose that the cofilin-induced change of helicity produced locally on one filament can be transmitted to the adjacent filaments within the bundle (Fig. 6D).”

So far, we have been unable to propose alternative mechanisms that could explain our observations in light of what is known for cofilin at the single filament level (a similar point was raised by reviewer #2, please see above).

Areas within the paper, if addressed, will improve the arguments presented as well as the readability of the paper.

(1) The authors use both the terms cofilin binding (in section I of the results) as well as cofilin nucleation (in section III of the results). It is unclear if these terms are meant to indicate the same, or different, processes. The manuscript would benefit from a clear explanation of the steps of cofilin-mediated disassembly measured and quantified in the experiments, namely nucleation (or binding), cluster growth, and filament or bundle fragmentation. A clear description of these steps would also allow the reader to follow the logic of the experiments from Figure 3 to Figure 5.

We have edited the introduction to better describe the different steps of cofilin activity, and to remove any ambiguity whereas we are referring to cofilin binding or cofilin nucleation.

(2) Throughout the paper, the authors move from single filaments, to 2-filament bundles, to multifilament bundles, using different concentrations of fascin and cofilin. Given the biphasic behavior of cofilin, namely that low concentrations favor severing and high concentrations can favor coating and filament stabilization, I think it is important that concentrations for the components are consistent across experiments, and if changes of concentrations of important components (such as cofilin and fascin) are changed, a clear explanation as to why is included.

As explained in the beginning of the result section, most of our experiments and quantification of cofilin activity using the microfluidics assay were done using 200 nM fascin and 200 nM cofilin as a standard. This is the case, in particular, for all the data shown in Fig 2, 3 and 4, where we compare the behavior of single filaments, 2-filament bundles, and larger bundles, exposed to the same protein concentrations.

We have also explored higher fascin and cofilin concentrations to document their respective impact, always mentioning any change in concentration. We agree with the reviewer that cofilin activity is biphasic at the single filament level (in the range of 0 to 1 μ M for mammalian ADF/cofilin, at physiological pH 7.4). In the case of fascin-induced bundles (already for two-filament bundles), filament saturation by cofilin, and thus their stabilization, will occur at higher cofilin concentration. This is mainly due to the lower nucleation activity of cofilin on fascin-induced bundles, preventing the nucleation of numerous cofilin clusters that will eventually fuse together, thus preventing saturation of filament bundles by cofilin before bundle fragmentation.

(3) In Figure 2, it is mentioned that for the spectrin seeds with the microfluidics, the filaments consisting of larger bundles were not analyzed along with the single filament and 2-filament bundles. Instead, a different experiment with seeds attached to beads is used to assess larger filament bundles. Why were larger bundles not analyzed in the microfluidic experiment?

We appreciate the insightful observation by the reviewer. When elongating actin filaments from spectrin-actin seeds, the seeds are randomly located on the glass coverslip of the microfluidics chamber. Upon exposure to fascin, only a subsection of any filament will be in contact with one or multiple filaments, ultimately forming a bundle due to the presence of fascin. In the case of high filament densities leading to large bundles, it is very difficult to identify the exact subsection of each filament which is engaged in a bundle or not. Despite our attempts to image individual filaments before and after exposure to fascin for enhanced clarity, the inherent difficulty persisted.

This limitation hindered our ability to quantify cofilin activity on large bundles when using spectrin-actin seeds randomly distributed on glass. To address this, we opted for an alternative approach involving micron-sized beads coated with spectrin-actin seeds. This modification not only circumvents the aforementioned limitation but also aids in the formation of larger bundles (up to 10 filaments per bundle). This adjustment significantly enhances our ability to study and quantify cofilin activity on larger bundles, contributing to a more robust and comprehensive understanding of cofilin activity on bundles.

And conversely, why were 2 filament bundles not assessed with the beads? Comparing the findings on two filament bundles with the findings on multifilament bundles would be easier for the reader if the small and large bundles were evaluated in the same experiments. If this is not experimentally feasible, the authors need to provide clearer explanation as to why this analysis is not included.

Actually, we did assess 2-filament bundles in the bead assay. The cofilin activity on 2-filament bundles from beads are reported, along with larger bundles, in figure 3E-F for nucleation, and in figure 4C for cofilin cluster growth rates.

(4) The authors indicate that at increased fascin concentration (1 μM) that single filaments decrease the nucleation rate of cofilin clusters. The authors should comment on the mechanism for fascin (at 1 μM concentration) for affecting cofilin binding.

We thank the review for this comment. We now comment on this mechanism in the result section: "This observation is consistent with the low affinity of fascin for the side of single actin filaments. Furthermore, this indicates that cofilin and fascin may have overlapping binding sites, or that a more complex competition may exist between the two proteins, where the binding of one protein would induce conformational changes on neighboring actin subunits affecting the binding of the other protein."

(5) The authors should determine and include the dissociation rate for the labeled cofilin used in this study, especially given the proposed mechanism for cofilin excluding fascin within the bundles.

- If the reviewer means that we need to characterize the behavior of the labeled cofilin: in Wioland et al 2017, we have previously reported that cofilin dissociates slowly from cluster boundaries (at 0.7 s^{-1} for cofilin-1 on alpha-skeletal rabbit actin, as used in the present study) and extremely slowly from inside a cofilin cluster ($\sim 2 \cdot 10^{-5} \text{ s}^{-1}$).

- If the reviewer means that we should investigate the competition between fascin and cofilin along bundles: we agree that this is indeed an interesting question. However this is quite complex because many unknown parameters are involved. In addition to the on/off-rates of each protein and how it is affected by the presence or the proximity of the other protein, we need to consider that fascin has fewer binding sites than cofilin, and that their accessibility changes as the helicity of the filament evolves as cofilin binds. Investigating this question would require many experiments, which we would need to confront to a model. We believe that this is out of the scope of this manuscript.

(6) For Figure 4, D and E, what do the dynamics of fascin and cofilin signal look like on a larger filament bundle? It would be informative to provide the cofilin cluster nucleation rate on larger filament bundles with a range of fascin concentrations (as in 3D for a two filament bundle). It would be interesting indeed to investigate the dynamics of fascin and cofilin on larger bundles. However, this experiment is quite challenging due to the fluorescence background of fluorescently-labeled fascin in our microfluidics assay (regardless of bundle size). We have been unable to perform this assay with success on large bundles. Moreover, it is difficult for us to carry out more of these experiments now that the first author of the study has left the lab. However, based on our results, we would expect that, for large bundles, increasing fascin concentration would also have a limited impact on the reduction of cofilin nucleation. Indeed, for 2-filament bundles, we can note that the increase of fascin concentration has a more limited impact on the nucleation of cofilin clusters (fig. 3D, roughly ~ 2 fold decrease for fascin from 100 to 500 nM), than the number of filaments per bundle (fig. 3F, a 10-fold decrease when increasing the size of a bundle from 2 to 10 filaments).

(7) Additionally, it would be useful to report the cofilin severing rate at a range of cofilin concentrations, at least for the 2 filament bundles.

Cofilin severing rate is not dependent on cofilin concentration in solution. This has been reported previously by several groups, including ours (e.g. Suarez et al, Current Biology 2011 ; Gressin et al, Current Biology 2015; Wioland et al, Current Biology 2017).

Below is the comparison of cofilin cluster severing at 100 and 200 nM cofilin, on single actin filaments, which we added to supplementary figure 10.

At 100 nM cofilin, we measured a similar cofilin cluster severing rate on 2-filament bundles, by measuring the survival fraction of overlapping cofilin clusters that lead to 2-filament bundle fragmentation over time. The figure pasted below is new Supp. Fig. 11.

When the severing occurs in the two filament bundles, does this severing occur mostly at boundaries with cofilin-actin and bare actin or does this severing occur at cofilin-actin/fascin-actin boundaries?

This is an interesting point. In the presence of a saturating amount of fascin, on 2-filament bundles, one fascin protein is bound every 13 actin subunits along each filament of a bundle. Most of the time, a cofilin boundary will not be in contact with a fascin-bound actin subunit. The limited spatial resolution of optical microscopy does not allow to say whether fascin was present at the boundary of a cofilin cluster or not when severing occurred. Nonetheless, we show that cofilin cluster severing is unaffected by fascin-bundling (i.e. severing rates per cofilin cluster boundary are similar on single filaments and on 2-filament bundles). Overall, bundling by fascin probably does not change the way cofilin severs, i.e. it occurs at the boundary between cofilin-decorated and bare actin regions.

(8) For the images of large bundles appearing braided in figure 6A, the lower left panel the braided appearance is not obvious. Additionally, what is the number of filaments in the bundles shown? Finally, given that in Figure 3F it is indicated that cofilin cluster nucleation events are rare on large bundles, and the cluster growth rate is reduced on large bundles (Figure 4C), the authors need to indicate how frequently this braided appearance is observed as well as what the nucleation rate, growth rate and severing rate is for 500nM cofilin on bundles.

We have repeated the negative staining EM experiments several times and now show new images and quantification (new Supp. Fig. 13). In our new series of experiments, the braiding that was previously shown in Fig. 6 proved difficult to reproduce and to quantify. We therefore decided to remove EM observations from the main fig 6, and we no longer present them as evidence supporting the mechanism that we propose for inter-filament cooperativity.

As stated in point (7) above, the severing rate is independent of cofilin concentration. We've used 500 nM cofilin, which is a rather high cofilin concentration, to investigate bundle fragmentation in EM, as in solution we mostly form large bundles and they are more slowly targeted by cofilin than individual or 2-filament bundles (figure 3F & 4C). At the single filament and 2-filament bundle level, the nucleation of cofilin clusters is extremely fast at 500 nM cofilin ($> 10^{-4} \text{ s}^{-1}$ per binding site).

(9) The authors indicate that the rapid fragmentation of twist constrained 2-filament bundles prevented them from directly quantifying the nucleation rate of the subsequent cofilin clusters that overlapped the initial ones. I'm unclear why this is the case, and if this is the case, I don't understand how the authors can be sure that a second nucleation event occurred in the twist constrained bundles. From the experimental data in 7C, it appears that the fragmentation rate for two filament bundles is similar to the fragmentation rate for twist constrained single filaments. The authors need to clearly state what they were able to observe and quantify as well as include

the timing for this severing. If the authors could not observe a second nucleation event prior to severing, this should be clearly stated.

Fragmentation of a 2-filament bundle requires the severing of two co-localized cofilin clusters, one on each filament. When 2-filament bundles are twist-constrained the sequence of events leading to bundle fragmentation is fast, thus it is difficult to separate the events within the resolution of our experiment. In this case, cofilin clusters sever quickly, thus the size of the clusters is small, which translates into a low fluorescence intensity. Therefore, the quantification of the increase of cofilin fluorescence intensity along a bundle did not allow us to unambiguously identify the 'cooperative' nucleation of two-overlapping cofilin clusters before the bundle is fragmented. So, apart from the quantification of the nucleation of cofilin clusters, which we show is unaffected by twist-constraining the bundles, we were unable to measure the growth rate nor the severing rate of cofilin clusters.

Numerical simulations, using similar severing rates for cofilin clusters on both twist-constrained single filaments and 2-filament bundles, satisfactorily reproduce our experimental observations (dashed lines in Fig. 3C).

We have edited the 'Twist-constrained bundle fragmentation' section to clearly say what we measured and what could not be measured : "We observed that the nucleation rate of cofilin clusters was similar for both twist-constrained and twist-unconstrained fascin bundles (Supp. Fig. 15), in agreement with observations on single actin filaments (Wioland et al, 2019b).

The rapid fragmentation of twist-constrained 2-filament bundles prevented us from directly quantifying the nucleation rate of the subsequent cofilin clusters that overlapped with the initial ones, as well as cluster growth and severing rates."

This could be due to the rapid fragmentation, but it could also be due to severing occurring in the absence of a second cofilin nucleation event. It would be informative to compare the time from cofilin nucleation to severing event for two filament bundles in twist constrained and unconstrained. Clarification of the dynamics of nucleation and spreading of cofilin and the timing of fragmentation of the twist constrained filament bundles is needed.

As explained in the previous point, cofilin-induced severing occurs significantly faster on twist-constrained single actin filaments compared to unconstrained filaments.

For twist-unconstrained filament bundles, we never observed bundle fragmentation that originated from only one cofilin cluster. For twist-constrained bundles, while our observation is limited by the rapid fragmentation of the bundles, it is hard to imagine that a single cofilin cluster on one filament would induce the fragmentation of the neighboring filament. Recently, Bibeau et al, PNAS 2023, using magnetic tweezers to twist single actin filaments, showed that, without cofilin, applying up to 1 rotation/ μm to an actin filament does not cause its fragmentation. It is thus reasonable to say that cofilin binding is required to fragment twist-constrained filaments. Moreover, in our numerical simulations (without inter-filament cooperativity, faithfully reproducing the kinetic of 2-filament fragmentation observed in microfluidics), 75% of bundle fragmentation resulted from a sequential nucleation of cofilin clusters, with the nucleation of the second cofilin cluster occurring after the first cofilin cluster has already severed one filament of the bundle.

(10) Discussion of how twist constrained fragmentation dynamics might affect the dynamics of larger bundles in structures such as filopodia would be useful.

We had substantially edited the discussion section of the manuscript, attempting to better discuss the physiological implications of our in vitro observations (bundle size & twist-constraints).

Minor changes that would improve the paper:

(11) In Figure 1C, Figure 2B and Figure 2E, the indication, on the graph, of the fold-change between the rates is confusing as it is not clear from the labeling on the graph that the x15 is referring to the slope of the lines, keeping this information in the legend is appropriate, but if it is to be included on the graph, perhaps adding in the linear fit on the graph is also needed.

We have edited the figures accordingly, and included fit lines in figure 1.

(12) Figure 7A, lining up the diagram with the kymographs below would help improve interpretation of the diagram and simulation. Alternatively, if the diagram (upper) in A does not diagram the kymographs below, this needs to be clearly stated, and it would be preferable that the diagram above matches the kymographs below.

We have edited the figure layout accordingly.

(13) Despite referencing the Breitsprecher, 2011 paper in the introduction, the authors do not explain how their results showing that cofilin fragments filament bundles slower than single actin filaments correspond with the Breitsprecher findings that fascin bundles favors cofilin filament severing. While the authors do not need to explain the Breitsprecher data, if they reference these findings that run counter to their results, an explanation for the discrepancy would be reasonable to include in the discussion.

We agree with the reviewer comments, which was also a comment made by reviewer #2. We now more directly discuss possible discrepancies between Breitsprecher and our studies :
“Previously, using pyrene-actin bulk experiments, Breitsprecher and colleagues reported a diminished cofilin binding to fascin-induced filament bundles (Breitsprecher et al, 2011). In spite of this, their observation of fluorescently labeled actin filament bundles seemed to indicate an efficient severing activity. Since cofilin was not fluorescently labeled, they could not observe cofilin clusters, and they proposed that severing was enhanced because fascin served as anchors along filaments and impeded cofilin-induced changes in filament helicity. This proposed mechanism bears resemblance to our previously reported findings for artificially twist-constrained single actin filaments (Wioland et al, 2019b). Here, we show that this mechanism does not occur in fascin-induced bundles.”

Reviewer #3 (Significance (Required)):

The research presented in "Fascin-induced bundling protects actin filament from disassembly by cofilin" is relevant and of interest to the field as it directly addresses a limitation in our understanding of how cofilin-induced severing occurs in F-actin bundles. Bundled F-actin may constitute the majority of linear F-actin within the cell and is specifically important in F-actin-based structures such as filopodia and stress-fibers. The data supports a model for interfilament

Full Revision

cooperativity that provides a molecular mechanism for cofilin-mediated severing of fascin-bundled filaments.

February 2, 2024

RE: JCB Manuscript #202312106T

Dr. Antoine Jégou
Institut Jacques Monod
15 rue Hélène Brion
Paris 75013
France

Dear Dr. Jégou,

Thank you for submitting your revised manuscript entitled "Fascin-induced bundling protects actin filaments from disassembly by cofilin." The manuscript has now been re-reviewed by original Review Commons referees #2&3. We would be happy to publish your paper in JCB pending final text changes recommended by the reviewers as well as revisions necessary to meet our formatting guidelines (see details below).

A. MANUSCRIPT ORGANIZATION AND FORMATTING:

1) Text limits: Character count for Articles is < 40,000, not including spaces. Count includes title page, abstract, introduction, results, discussion, and acknowledgments. Count does not include materials and methods, figure legends, references, tables, or supplemental legends.

2) Figure formatting: Articles may have up to 10 main text figures. Scale bars must be present on all microscopy images, including inset magnifications. Molecular weight or nucleic acid size markers must be included on all gel electrophoresis. Please avoid pairing red and green for images and graphs to ensure legibility for color-blind readers. If red and green are paired for images, please ensure that the particular red and green hues used in micrographs are distinctive with any of the colorblind types. If not, please modify colors accordingly or provide separate images of the individual channels.

3) Statistical analysis: Error bars on graphic representations of numerical data must be clearly described in the figure legend. The number of independent data points (n) represented in a graph must be indicated in the legend. Please, indicate whether 'n' refers to technical or biological replicates (i.e. number of analyzed cells, samples or animals, number of independent experiments). If independent experiments with multiple biological replicates have been performed, we recommend using distribution-reproducibility SuperPlots (please see Lord et al., JCB 2020) to better display the distribution of the entire dataset, and report statistics (such as means, error bars, and P values) that address the reproducibility of the findings.

Statistical methods should be explained in full in the materials and methods. For figures presenting pooled data the statistical measure should be defined in the figure legends. Please also be sure to indicate the statistical tests used in each of your experiments (both in the figure legend itself and in a separate methods section) as well as the parameters of the test (for example, if you ran a t-test, please indicate if it was one- or two-sided, etc.). Also, if you used parametric tests, please indicate if the data distribution was tested for normality (and if so, how). If not, you must state something to the effect that "Data distribution was assumed to be normal but this was not formally tested."

4) Abstract: Please revise the abstract as requested by Reviewer #2.

5) Materials and methods: Should be comprehensive and not simply reference a previous publication for details on how an experiment was performed. Please provide full descriptions (at least in brief) in the text for readers who may not have access to referenced manuscripts. The text should not refer to methods "...as previously described" so please provide additional details in the Protein purification section.

6) For all cell lines, vectors, constructs/cDNAs, etc. - all genetic material: please include database / vendor ID (e.g., Addgene, ATCC, etc.) or if unavailable, please briefly describe their basic genetic features, even if described in other published work or gifted to you by other investigators (and provide references where appropriate). Please be sure to provide the sequences for all of your oligos: primers, si/shRNA, RNAi, gRNAs, etc. in the materials and methods. You must also indicate in the methods the source, species, and catalog numbers/vendor identifiers (where appropriate) for all of your antibodies, including secondary. If antibodies are not commercial, please add a reference citation if possible. Please indicate the catalog numbers for labeling

reagents such as esters and maleimides.

7) Microscope image acquisition: The following information must be provided about the acquisition and processing of images:

- a. Make and model of microscope
- b. Type, magnification, and numerical aperture of the objective lenses
- c. Temperature
- d. Imaging medium
- e. Fluorochromes
- f. Camera make and model
- g. Acquisition software
- h. Any software used for image processing subsequent to data acquisition. Please include details and types of operations involved (e.g., type of deconvolution, 3D reconstitutions, surface or volume rendering, gamma adjustments, etc.).

8) References: There is no limit to the number of references cited in a manuscript. References should be cited parenthetically in the text by author and year of publication. Abbreviate the names of journals according to PubMed.

9) Supplemental materials: Articles typically have up to 5 supplemental figures and 10 videos. You currently exceed this limit but, in this case, we will be able to give you the extra space but please try to consolidate some of these to reduce the overall number. You also have space for additional main figures so you may move some of this data into the main figures if you feel it is appropriate.

Please also note that tables, like figures, should be provided as individual, editable files. A summary of all supplemental material should appear at the end of the Materials and methods section. Please include one brief sentence per item.

10) Video legends: Should describe what is being shown, the cell type or tissue being viewed (including relevant cell treatments, concentration and duration, or transfection), the imaging method (e.g., time-lapse epifluorescence microscopy), what each color represents, how often frames were collected, the frames/second display rate, and the number of any figure that has related video stills or images.

11) eTOC summary: A ~40-50 word summary that describes the context and significance of the findings for a general readership should be included on the title page. The statement should be written in the present tense and refer to the work in the third person. It should begin with "First author name(s) et al..." to match our preferred style.

13) A separate author contribution section is required following the Acknowledgments in all research manuscripts. All authors should be mentioned and designated by their first and middle initials and full surnames. We encourage use of the CRediT nomenclature (<https://casrai.org/credit/>).

14) ORCID IDs: ORCID IDs are unique identifiers allowing researchers to create a record of their various scholarly contributions in a single place. Please note that ORCID IDs are required for all authors. At resubmission of your final files, please be sure to provide your ORCID ID and those of all co-authors.

15) Journal of Cell Biology now requires a data availability statement for all research article submissions. These statements will be published in the article directly above the Acknowledgments. The statement should address all data underlying the research presented in the manuscript. Please visit the JCB instructions for authors for guidelines and examples of statements at (<https://rupress.org/jcb/pages/editorial-policies#data-availability-statement>).

B. FINAL FILES:

****It is JCB policy that if requested, original data images must be made available to the editors. Failure to provide original images upon request will result in unavoidable delays in publication. Please ensure that you have access to all original data images prior to final submission.****

****The license to publish form must be signed before your manuscript can be sent to production. A link to the electronic license to publish form will be sent to the corresponding author only. Please take a moment to check your funder requirements before choosing the appropriate license.****

Thank you for this interesting contribution, we look forward to publishing your paper in Journal of Cell Biology.

Sincerely,

Pekka Lappalainen, PhD
Monitoring Editor
Journal of Cell Biology

Dan Simon, PhD
Scientific Editor
Journal of Cell Biology

Reviewer #2 (Comments to the Authors (Required)):

Chikkireddy et al describe very interesting results that imply that fascin bundles are much less prone to cofilin-mediated severing than free filaments. All these results are exclusively based on elegant in vitro results, most of which I find totally fine and understandable. I already like the paper very much in the first version, and I have appreciated the removal of more difficult to explain EM from the main manuscript body, which has made the majority of the results more convincing.

I do agree with reviewer 3 though that a few more experiments on larger bundles would have been nice, simply because essentially all implications for bundles in situ will be relevant to larger than two-filament bundles, at least if speaking of microspikes or filopodia, correct? Therefore, most of the arguments on the implications for such bundles (see discussion) have to remain a little bit vague and hypothetical (such as the limited diffusion of cofilin molecules into the bundle). On a side note, I do not think the argument that the first author has left the lab is a valid one (see comment 6 by the authors to reviewer 3 in the rebuttal letter) to refrain from adding additional experiments, as it should always be possible to compensate for such changes with additional coworkers. However, I will not insist on more experiments being done in this direction if reviewer 3 is satisfied.

Instead, I have a few minor comments on text and wording and overall presentation of the results that the authors might find worth considering when preparing a revised, perhaps final manuscript version.

Minor comments:

1) I have in the meantime gotten the chance to inspect the three Supp movies after being provided by the authors upon request. For movie 1, it would be nice if the legend explained why the concentration of actin was reduced from 1 μ M to 0.15 μ M in the presence of cofilin. This is not specified. Furthermore, it would be more intuitive to write actin in yellow font (in analogy to cofilin in purple) and leave fascin in white font as it was unlabeled as far as I understand. For movies 2 and 3, it would be nice if actin and cofilin were labelled in the movies in analogy to movie 1, as this is missing. Furthermore, the legend to movie 2 currently reads: "Single actin filament bundles exposed to cofilin". However, the actin structures in Movie 2 look more like individual filaments to me, so I guess the term "bundle" in the title is inappropriate and should be corrected. In case of movie 3, it would be nice to know at which time point after cofilin addition the movie starts, as the cofilin is apparently already present from the very beginning. This could also simply be specified in the movie legend.

2) The previous abstract was partially amended in the revised manuscript version, but it has actually not improved. Specifically, the long, newly-added, second-last sentence starting with "Inter-filament cooperativity accelerates..." (lines 23-25) is very

complicated, and harbors a change of grammar in the last part that makes it very clumsy to read. I would recommend to simplify the sentence, or best split it into two sentences. In the last sentence (line 26), the correct term is "tuning actin network turnover", not networkS, which is grammatically incorrect in my view. On line 55 in the intro, "details" should be detail - plural not needed here.

3) In the intro (lines 71-72), the authors word "...are observed in filopodia and microspikes that emerge from the front of lamellipodia..." This wording is odd and should be corrected, since filopodia can emerge from the cell periphery independently of lamellipodia (even in their absence, and certainly not from their front), whereas structures that are commonly called microspikes are largely embedded into lamellipodia (so part of them) rather than emerging from their fronts.

4) The sentence in lines 667-8 also harbors a bit strange grammar with several subject changes, so I would suggest to reword it to: "... alpha-actinin, which forms dimers and thus constitutes a larger and more flexible crosslinker than fascin, may differentially impact..."

5) The final point has occurred to me after reading the comments by reviewer 3 and looking at the manuscript again. This concerns the issue that the authors here introduce for the first time (at least to my knowledge) the term "cofilin cluster NUCLEATION". I believe that this is a bit tricky, in particular because cofilin has been previously proposed, at least in vitro, to be capable of driving actin nucleation itself, at least at high concentrations, see e.g. Andrianantoandro et al., 2006 (PMID: 17018289). Whether or not this will be relevant in any in vivo setting, has remained a matter of debate I believe. However, in this respect, there are text sections in the current manuscript where readers might get a bit confused, e.g. line 241, in which the authors word: "This strong reduction in the nucleation rate thus appears to play a key role in protecting bundles from cofilin-induced fragmentation". So reading the sentence in isolation, it is difficult to understand initially whether the authors talk about cofilin cluster or actin nucleation.

So in light of all this, I have wondered why it is necessary to use a term as specific as nucleation. In case of actin, nucleation is very precisely defined as 3 monomers coming together to form a nucleus, and it is thus mechanistically quite easily distinguished from further elongation of a filament once a nucleus has formed. I am not convinced that the distinct nucleation of a cofilin cluster on filaments and its further growth are well-enough defined to be mechanistically separated as in case of actin filaments. So frankly, is it justified to use such a specific term really, or wouldn't it be better to replace the term nucleation by a more general one, such as cluster FORMATION or cluster INITIATION or alike? I think this could help to remove ambiguity between actin and cofilin cluster nucleation. In this case, cluster formation or cluster initiation could still be semantically distinguished from further cluster growth. Anyway, might be worth thinking about, also because reviewer 3 wondered about the term cofilin nucleation, and whether this was distinct from its simple binding.

Reviewer #3 (Comments to the Authors (Required)):

The research presented in "Fascin-induced bundling protects actin filament from disassembly by cofilin" is of great interest to the field of cell biology as it addresses a limitation in our understanding of how cofilin-induced severing occurs in F-actin bundles. The authors incorporate both rigorous in vitro evaluation of the dynamics of cofilin-mediated severing of fascin-bundled actin filaments along with simulations of these dynamics to establish a model for interfilament cooperativity that provides a molecular mechanism for cofilin-mediated severing of fascin-bundled filaments.

My only major concern regarding the manuscript was how the authors discussed the mechanism for interfilament cooperativity in cofilin binding. The authors modified their discussion to clarify that the mechanism described is a proposed model that strongly agrees with their reported results. The authors also improved the discussion with a concise summary of their findings which greatly improved the readability of the paper:

"Indeed, we report here the exclusion of fascin from within cofilin clusters, and a strong
573 increase in the nucleation of cofilin clusters on adjacent filaments. This inter-filament cooperativity
574 mechanism leads to the co-localized nucleation of cofilin clusters, and permits bundle
575 fragmentation faster than if the nucleation of cofilin clusters on adjacent filaments were purely
576 random. To our knowledge, this is the first time such inter-filament cooperativity is ever reported.
577 To explain this mechanism, we propose that the cofilin-induced change of helicity produced locally
578 on one filament can be transmitted to the adjacent filaments within the bundle (Fig. 6D)."

All the other minor points mentioned that might improve the manuscript were either addressed appropriately by the authors or are reasonably outside the scope of this manuscript (points 5 and 6). Point 8, regarding the negative staining EM, has been improved by quantifying the results and reporting this finding in a supplemental figure rather than a main figure in the manuscript.

Given the changes that the authors have made to the manuscript in light of the reviewers' comments, an already very strong and compelling paper of great interest to the field has been improved and I have no outstanding concerns about publication of this manuscript.

Antoine Jégou
Responsable de l'équipe
"Régulation de la dynamique d'assemblage de l'actine"
Institut Jacques Monod
15 rue Hélène Brion, 75013 Paris, France.
Téléphone : +33.(0)1.57.27.80.13
antoine.jegou@ijm.fr

Manuscript number: 202312106T
Corresponding author(s): Antoine, Jégou

Paris, February 22nd, 2024

Dear editor,

We are submitting a fully revised version of our manuscript in light of the points raised by the reviewers after the first round of revisions at Review Commons. We have edited the manuscript (changes are highlighted in red in the text), prepared the figures and videos, and addressed all the points raised by the two reviewers. Please see below.

Reviewer #2 (Comments to the Authors (Required)):

Chikkireddy et al describe very interesting results that imply that fascin bundles are much less prone to cofilin-mediated severing than free filaments. All these results are exclusively based on elegant in vitro results, most of which I find totally fine and understandable. I already like the paper very much in the first version, and I have appreciated the removal of more difficult to explain EM from the main manuscript body, which has made the majority of the results more convincing.

I do agree with reviewer 3 though that a few more experiments on larger bundles would have been nice, simply because essentially all implications for bundles in situ will be relevant to larger than two-filament bundles, at least if speaking of microspikes or filopodia, correct? Therefore, most of the arguments on the implications for such bundles (see discussion) have to remain a little bit vague and hypothetical (such as the limited diffusion of cofilin molecules into the bundle). On a side note, I do not think the argument that the first author has left the lab is a valid one (see comment 6 by the authors to reviewer 3 in the rebuttal letter) to refrain from adding additional experiments, as it should always be possible to compensate for such changes with additional coworkers. However, I will not insist on more experiments being done in this direction if reviewer 3 is satisfied.

Instead, I have a few minor comments on text and wording and overall presentation of the results that the authors might find worth considering when preparing a revised, perhaps final manuscript version.

Minor comments:

1) I have in the meantime gotten the chance to inspect the three Supp movies after being provided by the authors upon request. For movie 1, it would be nice if the legend explained why the concentration of actin was reduced from $1\mu\text{M}$ to $0.15\mu\text{M}$ in the presence of cofilin. This is not specified. Furthermore, it would be more intuitive to write actin in yellow font (in analogy to cofilin in purple) and leave fascin in white font as it was unlabeled as far as I understand. For movies 2 and 3, it would be nice if actin and cofilin were labelled in the movies in analogy to movie 1, as this is missing. Furthermore, the legend to movie 2 currently reads: "Single actin filament bundles exposed to cofilin". However, the actin structures in Movie 2 look more like individual filaments to me, so I guess the term "bundle" in the title is inappropriate and should be corrected. In case of movie 3, it would be nice to know at which time point after cofilin addition the movie starts, as the cofilin is apparently already present from the very beginning. This could also simply be specified in the movie legend.

We have corrected the legend of the movies and added overlays to the movies to better describe the experimental conditions.

2) The previous abstract was partially amended in the revised manuscript version, but it has actually not improved. Specifically, the long, newly-added, second-last sentence starting with "Inter-filament cooperativity accelerates..." (lines 23-25) is very complicated, and harbors a change of grammar in the last part that makes it very clumsy to read. I would recommend to simplify the sentence, or best split it into two sentences. In the last sentence (line 26), the correct term is "tuning actin network turnover", not networkS, which is grammatically incorrect in my view. On line 55 in the intro, "details" should be detail - plural not needed here.

We have corrected and simplified the complex sentence to make it simpler to understand.

3) In the intro (lines 71-72), the authors word "...are observed in filopodia and microspikes that emerge from the front of lamellipodia..." This wording is odd and should be corrected, since filopodia can emerge from the cell periphery independently of lamellipodia (even in their absence, and certainly not from their front), whereas structures that are commonly called microspikes are largely embedded into lamellipodia (so part of them) rather than emerging from their fronts.

We have corrected this sentence.

4) The sentence in lines 667-8 also harbors a bit strange grammar with several subject changes, so I would suggest to reword it to: "... alpha-actinin, which forms dimers and thus constitutes a larger and more flexible crosslinker than fascin, may differentially impact..."

We have reworded the sentence, following the reviewer suggestion.

5) The final point has occurred to me after reading the comments by reviewer 3 and looking at the manuscript again. This concerns the issue that the authors here introduce for the first time (at least to my knowledge) the term "cofilin cluster NUCLEATION". I believe that this is a bit tricky, in particular because cofilin has been previously proposed, at least in vitro, to be capable of driving actin nucleation itself, at least at high concentrations, see e.g. Andrianantoandro et al., 2006 (PMID: 17018289).

Whether or not this will be relevant in any in vivo setting, has remained a matter of debate I believe. However, in this respect, there are text sections in the current manuscript where readers might get a bit confused, e.g. line 241, in which the authors word: "This strong reduction in the nucleation rate thus appears to play a key role in protecting bundles from cofilin-induced fragmentation". So reading the sentence in isolation, it is difficult to understand initially whether the authors talk about cofilin cluster or actin nucleation.

So in light of all this, I have wondered why it is necessary to use a term as specific as nucleation. In case of actin, nucleation is very precisely defined as 3 monomers coming together to form a nucleus, and it is thus mechanistically quite easily distinguished from further elongation of a filament once a nucleus has formed. I am not convinced that the distinct nucleation of a cofilin cluster on filaments and its further growth are well-enough defined to be mechanistically separated as in case of actin filaments. So frankly, is it justified to use such a specific term really, or wouldn't it be better to replace the term nucleation by a more general one, such as cluster FORMATION or cluster INITIATION or alike? I think this could help to remove ambiguity between actin and cofilin cluster nucleation. In this case, cluster formation or cluster initiation could still be semantically distinguished from further cluster growth. Anyway, might be worth thinking about, also because reviewer 3 wondered about the term cofilin nucleation, and whether this was distinct from its simple binding.

We agree with the reviewer that the term 'nucleation' can be sometimes ambiguous, and have therefore replaced it with 'initiation' when referring to cofilin clusters.

-
Reviewer #3 (Comments to the Authors (Required)):

The research presented in "Fascin-induced bundling protects actin filament from disassembly by cofilin" is of great interest to the field of cell biology as it addresses a limitation in our understanding of how cofilin-induced severing occurs in F-actin bundles. The authors incorporate both rigorous in vitro evaluation of the dynamics of cofilin-mediated severing of fascin-bundled actin filaments along with simulations of these dynamics to establish a model for interfilament cooperativity that provides a molecular mechanism for cofilin-mediated severing of fascin-bundled filaments.

My only major concern regarding the manuscript was how the authors discussed the mechanism for interfilament cooperativity in cofilin binding. The authors modified their discussion to clarify that the mechanism described is a proposed model that strongly agrees with their reported results. The authors also improved the discussion with a concise summary of their findings which greatly improved the readability of the paper:

"Indeed, we report here the exclusion of fascin from within cofilin clusters, and a strong
573 increase in the nucleation of cofilin clusters on adjacent filaments. This inter-filament cooperativity
574 mechanism leads to the co-localized nucleation of cofilin clusters, and permits bundle
575 fragmentation faster than if the nucleation of cofilin clusters on adjacent filaments were purely
576 random. To our knowledge, this is the first time such inter-filament cooperativity is ever reported.
577 To explain this mechanism, we propose that the cofilin-induced change of helicity produced locally

578 on one filament can be transmitted to the adjacent filaments within the bundle (Fig. 6D)."

All the other minor points mentioned that might improve the manuscript were either addressed appropriately by the authors or are reasonably outside the scope of this manuscript (points 5 and 6). Point 8, regarding the negative staining EM, has been improved by quantifying the results and reporting this finding in a supplemental figure rather than a main figure in the manuscript.

Given the changes that the authors have made to the manuscript in light of the reviewers' comments, an already very strong and compelling paper of great interest to the field has been improved and I have no outstanding concerns about publication of this manuscript.

Sincerely yours,

Antoine Jégou